# Autonomous methane seep site monitoring offshore Western Svalbard: Hourly to seasonal variability and associated oceanographic parameters

Knut Ola Dølven[1], Bénédicte Ferré[1], Anna Silyakova[1], Pär Jansson[2,*], Peter Linke[3,*], and Manuel Moser[1,*]

[1]Centre for Arctic Gas Hydrate, Environment, and Climate,UiT The Arctic University of Norway, 9019 Tromsø, Norway
[2]Multiconsult Kyst og Marin, 9013 Tromsø, Norway
[3]GEOMAR Helmholtz Centre for Ocean Research Kiel, 24148 Kiel, Germany
[*]These authors contributed equally to this work

**Correspondence:** Knut Ola Dølven (knut.o.dolven@uit.no)

**Abstract.** Improved quantification techniques of natural sources are needed to explain variations in atmospheric methane. In polar regions, high uncertainties in current estimates of methane release from the seabed remain. We present two unique 10 and 3 months long time-series of bottom water measurements of physical and chemical parameters from two autonomous ocean observatories deployed at separate intense seabed methane seep sites (91 and 246 m depth), offshore Western Svalbard from 2015 to 2016. Results show high short term (100-1000 nmol $L^{-1}$ within hours) and seasonal variation, as well as higher (2-7 times) methane concentrations compared to previous measurements. Rapid variability is explained by uneven distribution of seepage and changing ocean current directions. No overt influence of tidal hydrostatic pressure or water temperature variations on methane concentration was observed, but an observed negative correlation with temperature at the 246 m site fits with hypothesized seasonal blocking of lateral methane pathways in the sediments. Negative correlation between bottom water methane concentration/variability and wind forcing, concomitant with signs of weaker water column stratification indicates increased potential for methane release to the atmosphere in fall/winter. We present new information about short- and long-term methane variability and provide a preliminary constraint on the uncertainties that arise in methane inventory estimates from this variability.

## 1   Introduction

Unexplained changes in atmospheric methane ($CH_4$) mole fraction motivates research in understanding and quantifying non-anthropogenic sources (Saunois et al., 2020). The atmospheric forcing of $CH_4$ is particularly sensitive to changes in emission rates due to a high warming potential and short lifetime. Improved knowledge about atmospheric $CH_4$ fluxes is therefore crucial to constrain future climate projections (Pachauri and Meyer, 2014; Myhre et al., 2016b). These properties of atmospheric $CH_4$ also makes reduced anthropogenic $CH_4$ emissions a potential solution for rapid climate change mitigation (Saunois et al., 2016). A global effort to cut greenhouse gas emissions through international agreements is, however, dependent on precise estimates of sources and sinks to verify contributions from different nations.

Seabed seepage is considered a minor source of atmospheric $CH_4$, but with high uncertainty in current and predicted emission estimates (Saunois et al., 2016). Current estimates suggest a total contribution of 7 (5-10) Tg yr$^{-1}$(Etiope et al., 2019; Saunois et al., 2020), which is ∼1% of the total $CH_4$ emissions to the atmosphere. Methane is released from the seabed as free gas (bubbles) and dissolved gas in sediment pore water. Bubbles rise quickly towards the sea surface, but most $CH_4$ dissolves near the seafloor because of gas exchange across the bubble rims and bubble dissolution (McGinnis et al., 2006; Jansson et al., 2019a). Dissolved $CH_4$ is dispersed and advected by ocean currents (Silyakova et al., 2020) and is continuously transformed to carbon dioxide ($CO_2$) by bacterial aerobic oxidation (Hanson and Hanson, 1996; Reeburgh, 2007). These processes significantly limit the lifetime of $CH_4$ in the water column and the amount of $CH_4$ that can reach the atmosphere is highly dependent on the depth where the seepage occurs (McGinnis et al., 2006; Graves et al., 2015). Intense $CH_4$ seepage at shallow depths in coastal areas and on continental shelves is therefore the main potential source of seabed $CH_4$ to the atmosphere.

The shallow continental margins of the Arctic Ocean store large amounts of $CH_4$ as free gas, gas dissolved in pore water fluid, and gas hydrates (James et al., 2016; Ruppel and Kessler, 2017), i.e. clathrate structures composed of water trapped by hydrocarbon molecules formed and kept stable at low temperature and high pressure (Sloan, 1998). Increasing bottom water temperature has the potential to liberate methane from these reservoirs via various mechanisms, potentially resulting in a positive climate feedback loop (Westbrook et al., 2009; Shakhova et al., 2010; James et al., 2016).

Studies on $CH_4$ inventory, distribution and release in the Arctic Ocean are mainly based on research cruise data from late spring to early fall, when ice and weather conditions allow field work in the region (Gentz et al., 2014; Sahling et al., 2014; Mau et al., 2017), whereas winter data is sparse. Bottom water temperature (Westbrook et al., 2009; Reagan et al., 2011; Ferré et al., 2012; Braga et al., 2020), water mass origins (Steinle et al., 2015), micro-seismicity (Franek et al., 2017), and hydrostatic pressure (Linke et al., 2009; Römer et al., 2016) have all been proposed to be linked with sources and sinks of $CH_4$ in the water column. These processes act on a wide range of time-scales, from hours (e.g. hydrostatic pressure) to decades (bottom water temperature). Without a better understanding of the spatial and temporal variability of $CH_4$ in Arctic Seep sites, it is challenging to untangle these processes. Unconstrained local variability in $CH_4$ seepage and concentration also imposes a high degree of uncertainty on $CH_4$ inventory estimates (Saunois et al., 2020). The combination of climate sensitive $CH_4$ storages, vast shallow ocean regions and limited data availability highlight the need for more understanding of seabed $CH_4$ seepage on Arctic shelves.

To assess the aforementioned challenges, we have obtained, analyzed and compared two unique long term underwater multi-parameter time series from two seafloor observatories deployed at two intense $CH_4$ seep sites on the western Svalbard continental shelf (Figure 1) where no $CH_4$ measurements have previously been done in winter season. We combine high frequency physical (ocean currents, temperature, salinity, pressure) and chemical ($O_2$, $CO_2$, $CH_4$) data to perform hypothesis testing and provide new insights on $CH_4$ distribution, content, as well as variability on short (minutes) and long (seasonal) timescales and potential implications.

## 1.1 Regional Settings

Two observatories ($O_{91}$ and $O_{246}$) were deployed from June 2015 (CAGE 15-3 cruise) to May 2016 (CAGE 16-4 cruise) from R/V *Helmer Hanssen* at the inter-trough shelf region between Isfjorden and Kongsfjorden, west of Prins Karls Forland. The $O_{91}$ observatory was deployed at 91 m water depth on the continental shelf (78.561$^o$N, 10.142$^o$E) and the $O_{246}$ observatory was deployed at 246 m water depth further offshore close to the shelf break (78.655$^o$N, 9.433$^o$E, Figure 1).

Both sites were located in areas with thousands of previously mapped $CH_4$ gas seeps (e.g. Sahling et al. (2014); Veloso-Alarcón et al. (2019); Silyakova et al. (2020); this work, see Figure 1), often referred as "flares" due to the appearance of bubble streams in echo-sounder data. Nonetheless, atmospheric sampling in this region suggests that any emissions to the atmosphere are small (Platt et al., 2018). Gas accumulation at the $O_{246}$ seep site has been suggested to be a result of gas migration in permeable layers within the seabed from deeper free gas or hydrate reservoirs (Rajan et al., 2012; Sarkar et al., 2012; Veloso-Alarcón et al., 2019), while seepage at site $O_{91}$ has been attributed to thawing sub-sea permafrost due to ice sheet retreat at the end of the last glaciation (Sahling et al., 2014; Portnov et al., 2016). Water sampling have indicated high temporal variability with bottom water concentrations (average) changing from 200 nmol $L^{-1}$ within 1 week in July 2014 at $O_{91}$ (Myhre et al., 2016a) and $\sim$ 80 nmol $L^{-1}$ within 20 hours (two single point measurements) at $O_{246}$ in August 2010 (Gentz et al., 2014). A consistent pattern of decreasing concentrations from the sea floor to the sea surface at both sites (400 to <8 nmol $L^{-1}$ at $O_{91}$ (Myhre et al., 2016a)) and from to >500 to <20 nmol $L^{-1}$ at $O_{246}$ (Gentz et al., 2014)) has also been observed. Further offshore, continuous measurements from a towed fast-response underwater laser spectrometer also revealed very high spatial $CH_4$ variability (Jansson et al., 2019b).

The local water masses are characterized by exchange and convergence of warm, saline Atlantic water (e.g. defined by Temperature T>3$^o$C and Salinity $S_A$ >34.9, Swift and Aagaard (1981)) in the West Spitsbergen current and colder, fresher Arctic water (e.g. T<0$^o$C, 34.3<$S_A$ <34.8, Loeng (1991)) in the Coastal Current combined with seasonal cooling, ice formation, and freshwater input from land (Nilsen et al., 2016) (Figure 1). Local mixing rates can be strongly affected by synoptic scale weather systems, causing upwelling and disruption of the front between the two ocean currents (Saloranta and Svendsen, 2001; Cottier et al., 2007). Freshwater input in summer stratifies the water column, while cooling, storm activity and sea ice formation can facilitate vertical mixing in winter (Saloranta and Svendsen, 2001; Nilsen et al., 2016).

## 2 Methods

The "K-Lander" ocean observatories were designed to monitor $CH_4$ release and associated physical and chemical parameters in challenging environments (see Appendix A). A launcher equipped with camera and telemetry allowed for safe deployment at a site selected by visual control. Observatory $O_{91}$ recorded data from 2 July 2015 to 6 May 2016, while $O_{246}$ recorded data from 1 July until 3 October 2015, when data recording ceased due to an electrical malfunction.

Both observatories were equipped with an Acoustic Doppler Current Profiler (ADCP), a CTD with oxygen optode, and Contros HydroC $CO_2$ *II* and HydroC *Plus* $CH_4$ sensors (Figure A1a, details in Appendix B). The deployed HydroC $CH_4$, being a younger iteration of the sensor, rely on a Tunable Diode Laser Absorption Spectrometry (TDLAS) detector (rather than

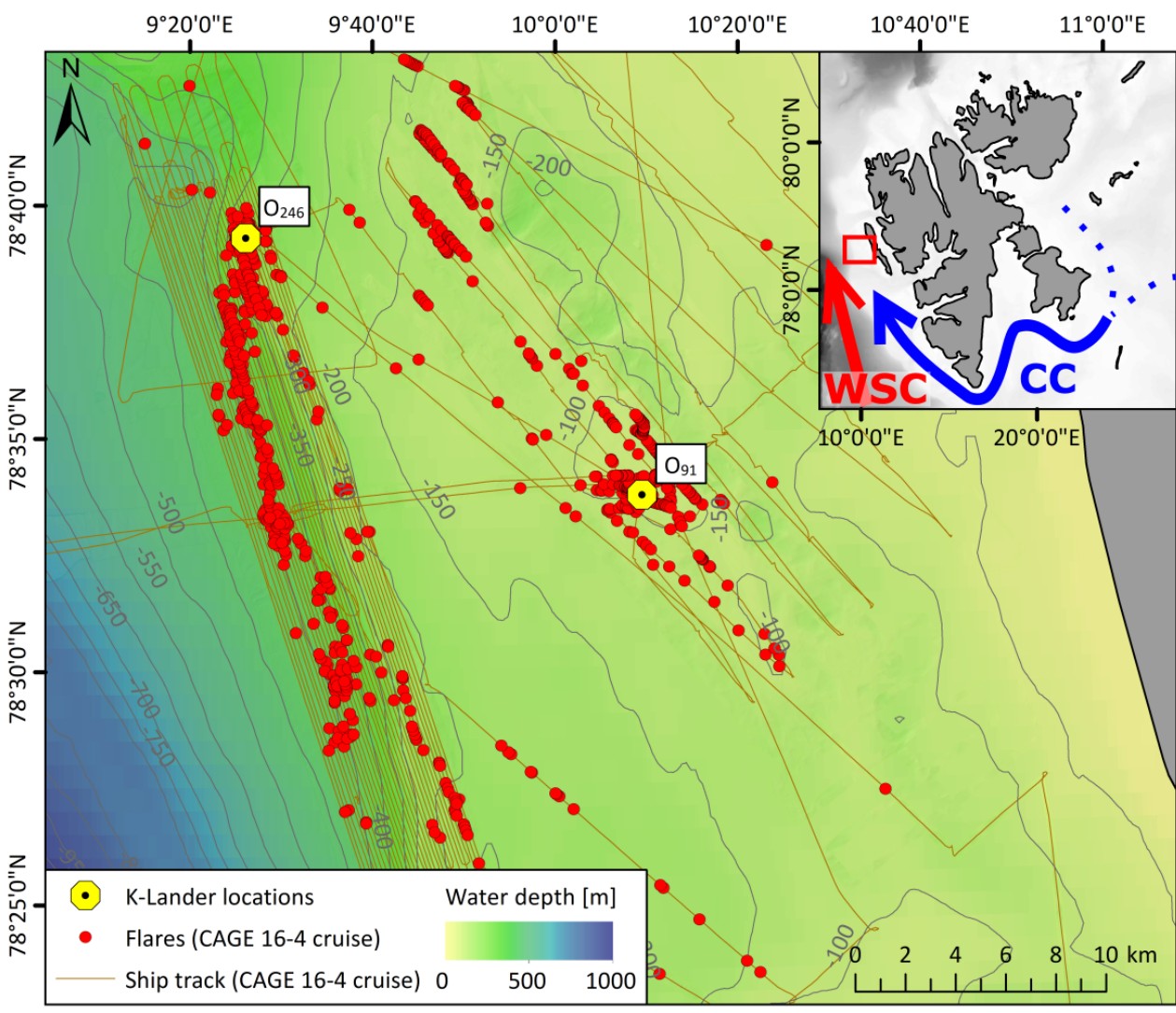

**Figure 1.** Bathymetry of the study area with location of the observatories $O_{91}$ and $O_{246}$ offshore western Svalbard. Flares detected by single-beam echo sounder survey prior to recovering the observatories (May 2016, cruise CAGE 16-4) are indicated with red dots and ship tracks as brown lines. The inset map shows the working area (red square) offshore Svalbard. WSC and CC refer to the warm West Spitsbergen Current and cold Coastal Current, respectively.

non-dispersive infrared spectrometry (NDIR)), while the $CO_2$ sensors use NDIR detectors. Both sensors were equipped with polydimethylsiloxate (PDMS) membranes, and a Seabird SBE 5M pump (see Appendix B).

High power consumption of the Contros HydroC $CH_4$ and $CO_2$ sensors required a power cycling mode to allow for long-term monitoring while simultaneously capturing rapid short-term variability. Partial pressure of $CH_4$ and $CO_2$ was therefore measured continuously for 24 hours every 21 days, and for one hour every day (see Table B1). Methane concentration data were corrected for slow response time following Dølven et al. (2021) onto a 3 minute interval grid and converted to absolute concentration, which is the default "$CH_4$ concentration" discussed and described in this text (see Appendix B). Faulty pumps in the $CO_2$ sensors ambiguously increased the response time which prevented response time correction, making $CO_2$ data suitable only for long-term qualitative analysis.

Uncertainty ranges for the $CH_4$ sensor data are reported as 95% confidence intervals and typically vary between 5 and 20% (Figure B1b). We did no post and/or intermittent validation. Although always an advantage for all sensors in long-term deployments, this is not a requirement for the TDLAS based sensor (as opposed to NDIR), due to its high long-term stability. Standard post-processing (e.g. inspection of meta data such as internal pressure and temperature) and evaluation of fit residuals in the response time correction procedure (see Appendix B and Dølven et al. (2021)) also indicated consistent sensor behavior throughout the deployments. It is also worth noting that the current manuscript concerns large changes and high concentrations and we are confident that the quality of the response time corrected Contros HydroC $CH_4$ data is sufficient to support the inferences described herein.

We calculated correlation coefficient (R) matrices to give a first order overview of linear relationships between the measured parameters. We mapped the flares in the area using single-beam echo-sounder data collected during the observatory recovery cruise in 2016 (CAGE 16-4, Figure 1) and estimated gas flow rates using the FlareHunter software (Veloso et al., 2015). Additionally, we obtained 10 m wind reanalysis data from the ERA-Interim database.

We calculated seawater density (McDougall and Barker, 2011) and $CH_4$ solubility (Kossel et al., 2013) using the CTD data. A CTD cast (SBE plus 24 Hz) prior to the $O_{91}$ recovery (6 May, 2016) showed a salinity drift in the conductivity sensor of around -0.4 (here and elsewhere in the paper, salinity values are practical salinity). Post-calibration, inspection of the conductivity signal and potential water mass mixing end-members indicates that this might have been caused by mud pollution occurring in late 2015 or early 2016.

## 3 Results

### 3.1 Time series at site $O_{91}$

Dissolved $CH_4$ concentration at site $O_{91}$ ranged from $5\pm3$ nmol $L^{-1}$ (6 December in 2015) to $1748\pm142$ nmol $L^{-1}$ (20 August in 2015) (Figure 2a and Appendix C), with 2.5 and 97.5 percentiles of 16 and 785 nmol $L^{-1}$. The data follows a nearly log-normal distribution, with a mean and median of 227 and 165 nmol $L^{-1}$, respectively, and interquartile range of 88-334 nmol $L^{-1}$. Large variations (>100 up to almost 1000 nmol $L^{-1}$) in $CH_4$ concentration occurred on short time-scales (<1 hour) throughout the measurement period (see Figure 2a, d, and all 24-hour periods in Appendix C) with an average range for all the

**Table 1.** Correlation coefficients between variables at $O_{91}$. "RTC $CH_4$" and "Raw $CH_4$" refers to response time corrected and untreated $CH_4$ data, respectively (Sect. 2 and Appendix B).

| | RTC $CH_4$ mol $L^{-1}$ | Raw $CH_4$ mol $L^{-1}$ | Temperature $^oC$ | Salinity | Oxygen mol $L^{-1}$ | Pressure dbar | Solubility mol $L^{-1}$ | Wind speed m $s^{-1}$ | $CO_2$ $\mu atm$ |
|---|---|---|---|---|---|---|---|---|---|
| RTC $CH_4$ | 1 | 0.91 | -0.06 | 0.23 | 0.03 | 0.08 | 0.06 | -0.33 | -0.25 |
| Raw $CH_4$ | 0.91 | 1 | -0.07 | 0.27 | 0.03 | 0.10 | 0.06 | -0.37 | -0.31 |
| Temperature | -0.06 | -0.07 | 1 | 0.69 | -0.94 | -0.01 | -0.99 | 0.37 | 0.29 |
| Salinity | 0.23 | 0.27 | 0.69 | 1 | -0.78 | -0.06 | -0.58 | 0.06 | 0.46 |
| Oxygen | 0.03 | 0.03 | -0.94 | -0.78 | 1 | 0.02 | 0.85 | -0.33 | -0.67 |
| Pressure | 0.08 | 0.10 | -0.01 | -0.06 | 0.02 | 1 | 0.16 | 0.00 | -0.10 |
| Solubility ($CH_4$) | 0.06 | 0.06 | -0.99 | -0.58 | 0.85 | 0.16 | 1 | -0.35 | -0.30 |
| Wind speed | -0.33 | -0.37 | 0.37 | 0.06 | -0.33 | 0.00 | -0.35 | 1 | 0.52 |
| $CO_2$ | -0.25 | -0.31 | 0.29 | 0.46 | -0.67 | -0.10 | -0.30 | 0.52 | 1 |

24-hour periods of 840 nmol $L^{-1}$ and median rate of change (ROC) of 3.2 nmol $L^{-1}$ min$^{-1}$. We also observe a long-term trend of decreasing running median (2-week window) concentrations towards winter, from 495 nmol $L^{-1}$ in July/August 2015 to 53 nmol $L^{-1}$ in January 2016 (Figure 2). There was a relatively weak, but significant negative correlation between the wind speed and $CH_4$ concentration ($R_{RTC}$=-0.33), but otherwise weak to non-existent linear relationships between $CH_4$ concentration and the measured ocean parameters (Table 1).

$CO_2$ averaged 403 $\mu$atm with an increase towards mid-November 2015 ($\sim$410 $\mu$atm) then a decrease until 6 May ($\sim$391 $\mu$atm) in 2016 (Figure 2a). $CO_2$ dropped to $\sim$305 $\mu$atm on 24 August, concurrent with a rapid decrease in salinity (-0.5), increase in temperature and oxygen, and high $CH_4$ concentration. The increase in oxygen rules out methanogenesis. Instead, there might be at least two explanations for the reduction of $CO_2$ and enrichment of $CH_4$: i) water column mixing brings oxygen-rich, warm and fresh surface water to deeper depth, and with it $CO_2$ depleted water or ii) methane enrichment by 130    zooplankton following the summer bloom.

Bottom water temperature increased steadily from $\sim$3 in July to $\sim$5.5 $^oC$ in October/November 2015, with occasional sharp shifts (T$\pm$1$^oC$) occurring within hours to days (Figure 2b). Temperature then decreased from the beginning of December to $\sim$1.8$^oC$ at the end of the deployment in May 2016, showing more frequent and stronger episodes of rapid temperature shifts (T$\pm$2$^oC$ also occurring on hours-days). Despite uncertainty in salinity data, it is worth noting that these rapid shifts in 135    temperature and salinity were reproduced by the Svalbard 800 model in the same area (Silyakova et al., 2020) by eddy activity.

Hydrostatic pressure was mostly governed by tides (94.5% of variance) with dominant semi-diurnal M2 tide (M2 refers to a tidal constituent with period 12.42 hours, see e.g. Gerkema (2019)). Amplitudes varied from $\sim$1.2 to 1.5 meter during neap and spring cycles (Figure 2c).

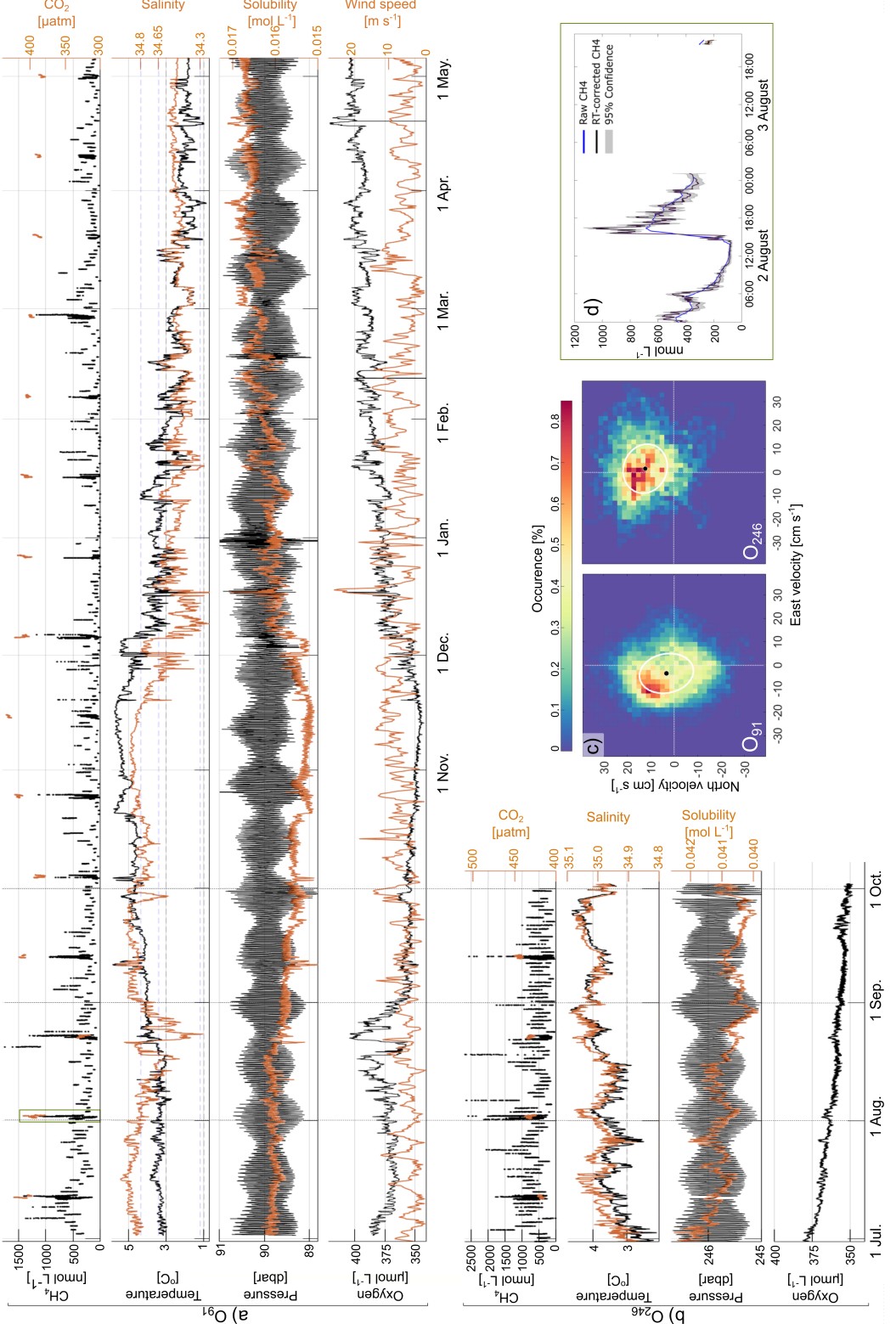

**Figure 2.** Time series from a) $O_{91}$ and b) $O_{246}$ showing response time corrected (see Appendix B) $CH_4$, $CO_2$, temperature, salinity, pressure, $CH_4$ solubility, oxygen, and wind speed (10 m) data. The $O_{246}$ data is truncated due to an electrical malfunction in the system on 3 October c) 2-d histogram and 1 standard deviation variance ellipse of bottom current velocity (81 m depth at $O_{91}$ and 236 m depth at $O_{246}$) and d) example of 24-hour and 1-hour (2 and 3 August) $CH_4$ concentration measurement period from $O_{91}$ (green box). All 24-hour measurement periods are shown in Appendix C. Note different scales between $O_{91}$ and $O_{246}$.

The calculated $CH_4$ solubility decreased from 0.016 mol $L^{-1}$ in July to 0.015 mol $L^{-1}$ in the end of November 2015, and increased to almost 0.017 mol $L^{-1}$ in May 2016 (Figure 2c). This long-term trend was mainly caused by temperature variability (R=-0.99), while tidal pressure changes caused a semi-diurnal variation of $\pm \sim 0.005$ mol $L^{-1}$.

Dissolved $O_2$ decreased from $\sim 385$ $\mu$mol $L^{-1}$ in July 2015 to $\sim 350$ $\mu$mol $L^{-1}$ at the beginning of December, and increased to $\sim 400$ $\mu$mol $L^{-1}$ towards 6 May, 2016 (Figure 2d) and followed temperature inversely (R=-0.94), with similar long and short-term variability.

The averaged bottom water current (81 m above the seafloor) was 4 cm $s^{-1}$ in a northwestward direction ($321^oN$) (Figure 2c). The current usually had one anti-clockwise rotation every 23.93 hour period, corresponding to the diurnal K1 tidal constituent (tide with period 23.93 hours, see Gerkema (2019)) with a secondary semi-diurnal (M2) modulation.

## 3.2    Time series at site $O_{246}$

$CH_4$ concentration at site $O_{246}$ ranged from $10\pm3$ nmol $L^{-1}$ on 21 September, 2015 to $2727\pm182$ nmol $L^{-1}$ on 18 August 2015, with 2.5 and 97.5 percentiles of 107 and 1374 nmol $L^{-1}$. The data approximately follows log-normal distribution with average and median of 577 and 600 nmol $L^{-1}$, respectively, and interquartile range of 293-721 nmol $L^{-1}$. The median RoC of $CH_4$ was almost 20 times higher compared to site $O_{91}$ with 31 nmol $L^{-1}$ $min^{-1}$ (Figure 2b and Appendix C). There was also clear diurnal periodicity in $CH_4$ concentration at $O_{246}$. The long-term trend (2-week running mean) shows decreasing concentrations until 3 October 2015 (end of the measuring period, Figure 2b). Dissolved $O_2$ decreased from $\sim 380$ $\mu$mol $L^{-1}$ to $\sim 300$ $\mu$mol $L^{-1}$ and was negatively correlated with water temperature (R=-0.61, see Table 2 for complete correlation matrix).

**Table 2.** Correlation coefficients between variables at $O_{91}$. "RTC $CH_4$" and "Raw $CH_4$" refers to response time corrected and untreated $CH_4$ (see Sect. 2 and Appendix B).

| | RTC $CH_4$ mol $L^{-1}$ | Raw $CH_4$ mol $L^{-1}$ | Temperature $^oC$ | Salinity | Oxygen mol $L^{-1}$ | Pressure dbar | Solubility mol $L^{-1}$ | Wind speed m $s^{-1}$ | $CO_2$ $\mu$atm |
|---|---|---|---|---|---|---|---|---|---|
| RTC $CH_4$ | 1 | 0.78 | -0.31 | -0.24 | 0.30 | 0.15 | 0.33 | -0.29 | -0.13 |
| Raw $CH_4$ | 0.78 | 1 | -0.45 | 0.26 | 0.48 | 0.10 | 0.45 | -0.44 | -0.09 |
| Temperature | -0.31 | -0.45 | 1 | 0.87 | -0.61 | -0.02 | -0.99 | 0.38 | 0.22 |
| Salinity | -0.24 | -0.26 | 0.87 | 1 | -0.22 | -0.03 | -0.87 | 0.07 | 0.13 |
| Oxygen | 0.30 | 0.48 | -0.61 | -0.22 | 1 | 0.06 | 0.59 | -0.65 | -0.41 |
| Pressure | 0.15 | 0.01 | -0.02 | -0.03 | 0.06 | 1 | 0.16 | -0.05 | 0.14 |
| Solu ($CH_4$) | 0.33 | 0.45 | -0.99 | -0.87 | 0.59 | 0.16 | 1 | 0.38 | -0.20 |
| Wind speed | -0.29 | -0.44 | 0.38 | 0.07 | -0.65 | -0.05 | 0.38 | 1 | 0.18 |
| $CO_2$ | -0.13 | -0.09 | 0.22 | 0.13 | -0.41 | 0.14 | -0.20 | 0.41 | 1 |

Temperature and salinity increased from ~2.5 to ~4.0 $^o$C and ~34.85 up to ~ 35.0, respectively, from the deployment until October 2015 (Figure 2b), with Atlantic water dominance throughout the measuring period. Rapid shifts of around $\pm1^o$C and 0.05 salinity occurred occasionally over a period of hours to days.

Variance in hydrostatic pressure was mainly explained by the tides (95.2%) which was mainly governed by the semi-diurnal M2 tide, with weaker diurnal and fortnightly modulation (Figure 2b). Changes in pressure varied from ~1.2 to ~1.5 m during periods of neap and spring tide.

Being governed mainly by temperature (R=-0.99), $CH_4$ solubility dropped from 0.042 mol L$^{-1}$ to 0.040 mol L$^{-1}$ from the deployment in July until October 2015, with a semi-diurnal variation of ~0.005 mol L$^{-1}$ due to tidal changes in hydrostatic pressure.

The averaged current was ~ 10 cm s$^{-1}$ northward (7$^o$N) (Figure 2c). Variability in the along-slope current (direction -10$^o$N) was strongly related to the semi-diurnal M2 tidal component, while the cross-slope currents were governed by the diurnal K1 frequency. The bottom water current rotated counterclockwise with a period of 23.93 hours (K1 tidal constituent), with semi-diurnal modulation in the along-slope component. Dissolved $CH_4$ concentration was weakly anti-correlated with wind speed (R=-0.29), temperature (R=-0.31), salinity (R=-0.24), and positively correlated with $CH_4$ solubility (R=0.33) and oxygen (R=0.3).

## 4 Discussion

### 4.1 $CH_4$ variability

Combining mapped flares and flow rates from the recovery cruise (May 2016) with bottom water current velocity (9 meters above the seafloor) reveals that $CH_4$ concentration was strongly affected by whether water was advected from areas where we mapped strong or weak seepage in May 2016 (Figure 3). Strong seeps (flow rate >200 mL$^{-1}$ min$^{-1}$) were mainly located between ~30 and 80 m to the north/northeast of site $O_{91}$ and only weak and more distant seepage was observed south-west of the observatory (Figure 3a). Consequently, averaged $CH_4$ concentration from water coming from north-east was ~440 nmol L$^{-1}$, while water from south-west averaged ~100 nmol L$^{-1}$. Similarly, a strong $CH_4$ seep (flow rate ~1200 mL min$^{-1}$) was mapped ~40 m north of site $O_{246}$, making water advected from this direction highly elevated in $CH_4$ with an average of ~1400 nmol L$^{-1}$ compared to the overall average of 577 nmol L$^{-1}$ (Figure 3b). The rapid changes in dissolved $CH_4$ can to a high degree be explained by this relationship, due to the high variability in ocean current velocity. That this relationship holds for most of the measuring period also shows that even though observed average concentration are lower in winter months, the seep configuration did not change significantly from July 2015 to May 2016 and dissolved $CH_4$ was efficiently dispersed in relatively high concentrations in the whole seepage area.

Furthermore, daily $CH_4$ concentrations at site $O_{91}$ were higher on average than the 24-hour measurements (313 vs. 200 nmolL$^{-1}$). This can be explained by the comparable measurement periodicity (24 hours) and tidal periodicity (23.93 hours) in the ocean currents, resulting in predominantly eastward advection during daily measurements, thus systematically transferring water from a weak seepage area (Figure 3). We did not observe this effect at site $O_{246}$, most likely due to less tidal variance

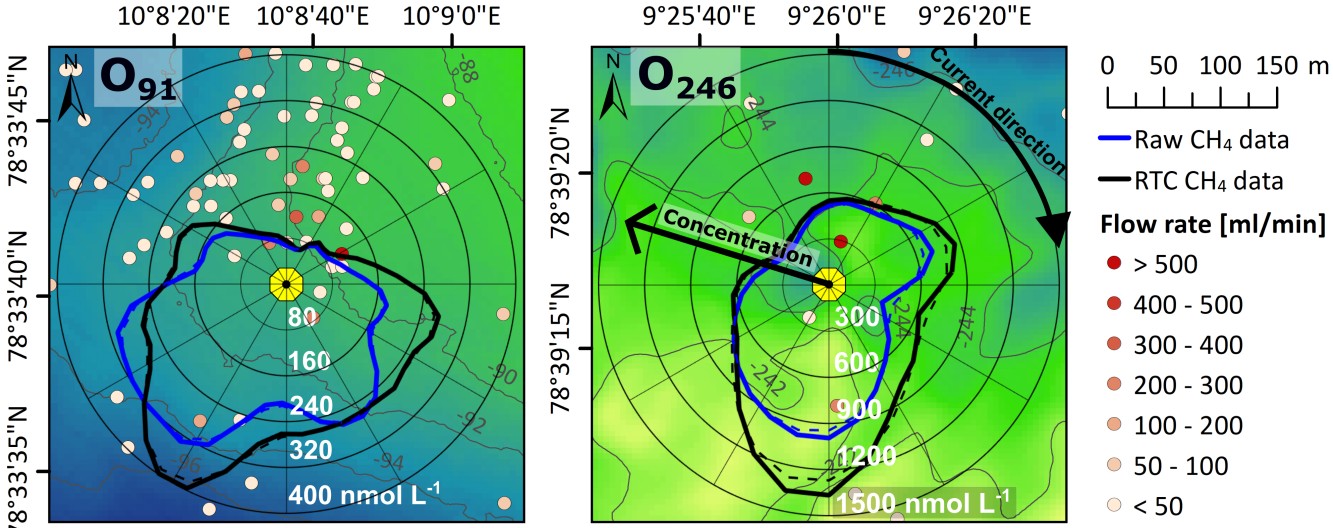

**Figure 3.** $O_{91}$ (left) and $O_{246}$ (right) location (yellow dot) as well as flow rates from flares mapped in its vicinity during CAGE 16-4 (colorscale). Background color (green-blue) illustrates seafloor bathymetry. Compass diagram show the relationship between ocean current direction (angle) and $CH_4$ concentration (distance from center, black is response time corrected (RTC) data and raw data is in blue).

in the current direction (Figure 2b). Nonetheless, this systematic tide-induced bias on the daily measurements at site $O_{91}$ highlights the importance of taking the oceanographic conditions into account to avoid misinterpretation of variability.

Since currents are mostly northward and seepage are mostly located to the north of both observatories, averaged measured $CH_4$ concentrations are likely lower than the average over the immediate surrounding area (Figure 3). Despite this, the observatory data show higher average $CH_4$ concentrations than previously reported. In the area surrounding site $O_{91}$, Silyakova et al. (2020) reported average concentration of 92, 70, and 61 nmol $L^{-1}$ in June 2014, July 2015, and May 2016, respectively, based on discrete water sampling. Averaged $CH_4$ concentrations measured at site $O_{91}$ in July 2015 and May 2016 were 566 and 110 nmol $L^{-1}$ respectively, i.e. around eight and two times higher than values reported by Silyakova et al. (2020). The maximum $CH_4$ concentration at $O_{91}$ of $1748 \pm 142$ nmol $L^{-1}$ on 20 August 2015 also significantly exceeds the previously maximum recorded concentration in the area of 480 nmol $L^{-1}$ (July 2014, Silyakova et al., 2020). At site $O_{246}$ the August 2016 average (564 nmol $L^{-1}$) was eight times higher than what Gentz et al. (2014) found in August 2010 (70 nmol $L^{-1}$), using an altimeter-controlled CTD towed at 2 meter above the seafloor. Maximum concentration in August 2016 also significantly exceeded previous observations, with $2661 \pm 163$ nmol $L^{-1}$ compared to 524 nmol $L^{-1}$ measured by Gentz et al. (2014).

These differences could be a result of temporal, local or regional differences in $CH_4$ concentration. However, strong vertical gradients in dissolved $CH_4$ are well documented at both seep sites (Gentz et al., 2014), and our sensors measured closer to the seafloor (1.2 m above seafloor), compared to Gentz et al. (2014) (2 m above seafloor) and Silyakova et al. (2020) (5 to 15 m above seafloor). Additionally, the observatories were deployed close to seeps using a launcher as opposed to "blind" water

sampling from ship-born rosette. Methane was also measured *in situ*, thereby avoiding potential $CH_4$ outgassing after retrieval of water samples (Schlüter et al., 1998).

Dissolved $CH_4$ within shallow seep sites where gas can bypass the oceanic sinks often present heterogeneous distribution and rapid temporal variability (Gentz et al., 2014; Myhre et al., 2016a). Our results show that the temporal variability at the two seep sites are higher than previously reported, and that changing ocean currents and configuration of nearby seeps are major contributors. This high short-term variability introduces a conceptual error in studies relying on discrete water sampling (e.g. to calculate inventories), since the time required to conduct the survey (∼days) is much longer than large temporal variations in concentration (up to order of $10^3$ nmol $L^{-1}$ within hours).

We can obtain a first order constraint on errors caused by short-term variability in a hypothetical water sampling survey using the 24-hour time-series from the observatories. We assume the hypothetical survey seeks to find the average concentration in the bottom layer of the seep site. The expected error can then be found by calculating the standard error of the mean (SEM) for a given number of samples $N$, using the 24-hour time-series as an underlying distribution representing the sub-daily variability of the seep site (Figure 4, Appendix D contains a detailed outline of the methodology). Even though surveys often require more than 24 hours to complete (2-3 days in Silyakova et al. (2020)), a majority of processes causing short-term variability have periods below or at ∼24 hours (for instance tides and many turbulent eddies see e.g. Sect. 3.2 and 3.1 and Talley et al. (2011)), likely making the daily distribution relevant also for surveys with longer duration. We compared SEM calculations based on the observatory 24-hour time-series with SEM calculations for the bottom water (∼5 meters above the seafloor) discrete water sample data used for average/inventory estimates of the $O_{91}$ seep site in Silyakova et al. (2020) (also included in Figure 4).

The absolute SEM (in nmol $L^{-1}$) is generally higher for time-series with higher averaged concentrations, making the relative SEM cluster well, with gradually diminishing range for increasing $N$ (an inherent property of the SEM, e.g. 12-45% for $N$=10, 9-30% for $N$=30 etc., Figure 4). The SEM of the data from Silyakova et al. (2020) is similar to the SEM of the 24-hour time-series, with a common range of 5-15% expected error for surveys with $N$ ∼60 samples ($N$=64,62, and 63 in Silyakova et al. 2020). It should be noted that the comparison with data from Silyakova et al. (2020) has caveats, e.g. that the observatory data does not contain errors due to spatial variability and an assumption of representative short-term temporal variability at the observatory sites (see also Appendix D).

Evidently, detailed surveys of individual seep sites, such as the study by Silyakova et al. (2020), can provide reasonable estimates of local inventories (<15% uncertainty) despite high short-term temporal variability. However, it is important to note that the area investigated in Silyakova et al. (2020) was densely mapped and homogeneous in the sense that it is an area where seepage is well documented (Silyakova et al., 2020). Interpolation or averaging across larger regions where the amount of seepage is mostly unknown can result in considerable errors due to false interpolation assumptions and amplification of individual measurement errors which can be large (expected errors up to ∼140% for single measurements, see listed standard deviations in Figure 4). These effects can potentially explain some of the discrepancies in estimates of oceanic $CH_4$ inventories and fluxes.

Our findings stress the importance of sufficiently dense mapping and knowledge about the underlying seep condition when collecting water samples for inventory estimates. They also highlight the advantage of towed or autonomous instrumentation

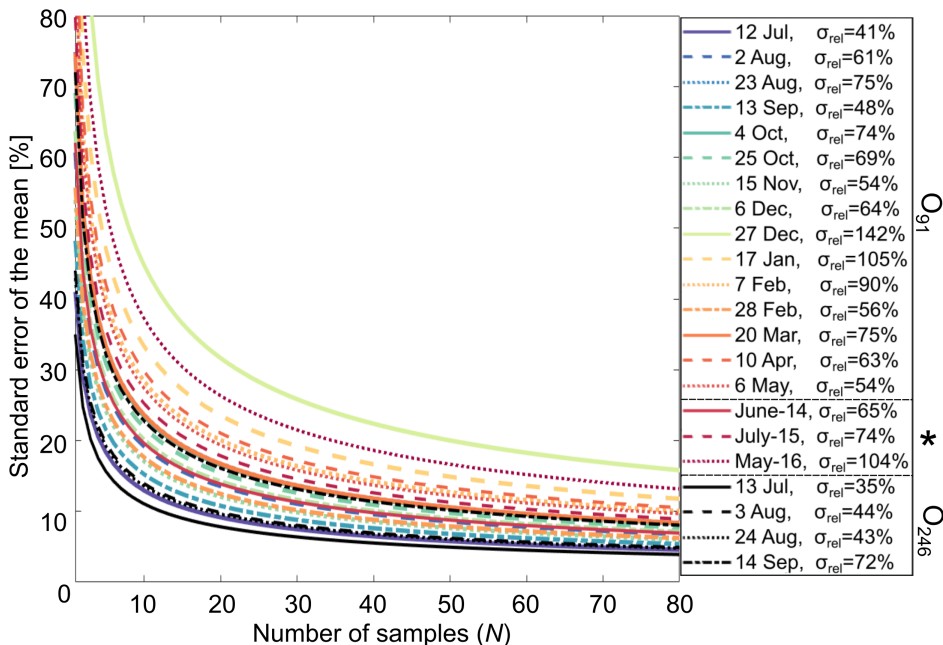

**Figure 4.** Relative standard error of the mean for different number of samples $N$ for $O_{91}$ 24-hour data, data presented in Silyakova et al. (2020) ("June-14","July-15","May-16"), and $O_{246}$ 24-hour data (in black color). Relative standard deviation (corresponding to the standard error with $N=1$) is given in the legend ($\sigma_{rel}$). * is data from Silyakova et al. (2020) calculated assuming that the sample distribution resembles the underlying distribution (see Appendix D).

capable of providing continuous $CH_4$ data, giving a considerably better coverage and representation of the $CH_4$ distribution in less time (e.g., Sommer et al., (2015); Grilli et al.(2018); Canning et al., (2021)). Assuming a distribution which better reflects the uneven spread of $CH_4$ when applying interpolation/extrapolation techniques could also limit estimation errors. Future studies should investigate how initial errors due to short-term and small scale variability propagate via different up-scaling
techniques and how these errors can be mitigated.

### 4.2 Hydrostatic pressure

Tidal changes in hydrostatic pressure can trigger $CH_4$ release by build-up of $CH_4$ in sediment pore-water at rising tide and subsequent release when pore pressure decreases at falling tide as observed at the Hikurangi Margin (Linke et al., 2009) and Clayoquot slope (Römer et al., 2016). Our study sites differ from these sites in depth (they are >600 m) and in tidal amplitude (4
250 m at Calyoquot slope compared to 1.5 offshore Prins Karls Forland). Linke et al., (2010) and Römer et al., (2016) also observed bubbles hydro-acoustically, while we measure dissolved $CH_4$ which is strongly affected by the (also tidally dependent) current direction (Figure 3).

To evaluate the effect that hydrostatic pressure changes have on the *in situ* concentration, we need to constrain the variance caused by changing current directions (since they operate in the same frequency domain). To do this, we first binned the $CH_4$

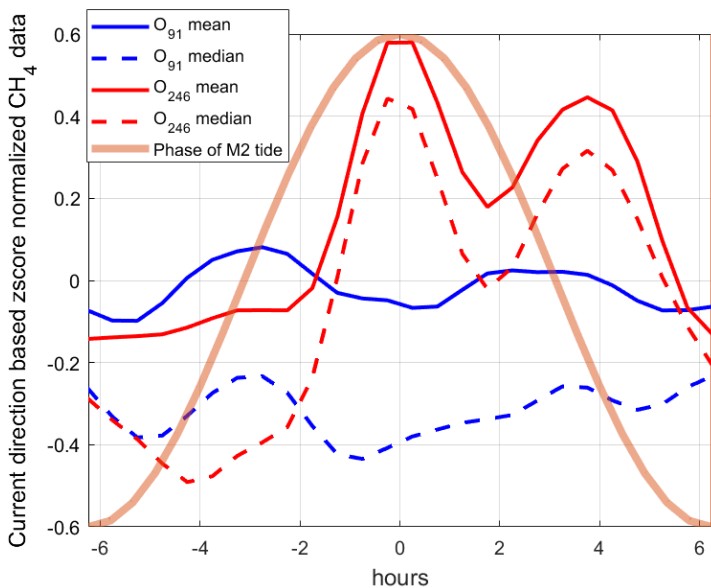

**Figure 5.** Median and averaged the standard scores of $CH_4$ binned according to bottom water current direction according to where the data were sampled on the phase of the M2 pressure tide.

concentration data into overlapping bins defined by the current direction at the time when the measurement was obtained and calculated standard scores (the number of standard deviations each value deviates from the sample mean, see e.g. Kreyszig (1979)) for the data in each bin. We used larger current direction intervals for $O_{246}$ due to the shorter data set, with a $12^o$ window for $O_{91}$ and a $30^o$ window for $O_{246}$. This resulted in a data set (i.e. the standard scores from all bins) effectively unrelated to the current direction. We then binned all the standard-scored $CH_4$ data according to when the data were collected

in relation to the M2 governed tidal cycle peak using overlapping 30 minute bins (the M2 tide explains 79.2% and 80.3% of the pressure variance at $O_{91}$ and $O_{246}$, respectively). Average and median values were calculated for each bin, giving the averaged/median normalized dissolved $CH_4$ value (standard score) for each current velocity defined data bin as a function of the M2 tidal cycle (Figure 5). This partial decoupling of variability in hydrostatic pressure and current direction was possible since the bottom water current and hydrostatic pressure changes had different dominant tidal constituents, i.e. the current was

mainly dominated by the diurnal K1 constituent ($\sim$23.91 hour period), while the M2 tide is semi-diurnal (12.42 hour period).

A strong effect of the hydrostatic pressure on local seepage should elevate the standard scores at decreasing pressure (from 0 to 6.2 hours, i.e. in the right half of Figure 5), which we observe at both observatories. However, we observe stronger peaks at increasing hydrostatic pressure (-3 hours) at site $O_{91}$ and at the M2 peak (0 hours) at site $O_{246}$, which contradicts this hypothesis. This does not mean that there is no effect of hydrostatic pressure changes, but rather that the seepage in the area is

widespread at both falling and rising tide conditions. The high variability caused by the strong effect of current direction also makes it particularly challenging to detect moderate changes in seepage intensity.

### 4.3 Bottom water temperature

Bottom water temperature can affect $CH_4$ release by altering hydrate stability and $CH_4$ solubility in pore water and water column (Sloan, 1998; Jansson et al., 2019a). Seasonal $CH_4$ release variability resulting from temperature variations in the bottom water has been linked to migration of the Gas Hydrate Stability Zone (GHSZ) and hydrate dissociation further offshore at $\sim 390$ m water depth (Berndt et al., 2014; Ferré et al., 2020). Our observatories were deployed in areas too shallow for gas hydrate to form. However, inversely varying seepage intensity between seepage at the GHSZ depth (390 m) and site $O_{246}$ can suggest that these areas are fed by the same hydrocarbon source and that hydrates seasonally block the lateral pathways between these seep sites (Veloso-Alarcón et al., 2019). This is in agreement with the observed long-term ($\sim 3$ months) negative correlation between bottom water temperature and dissolved $CH_4$ at site $O_{246}$ (R=-0.31). It should be noted that the same relationship is observed at $O_{91}$, however no geophysical data are available from this area due to the shallow depth.

Tidal pressure variations can affect $CH_4$ release via pore water solubility (Sect. 4.2), but on longer timescales, $CH_4$ solubility is almost exclusively a function of water temperature. Higher $CH_4$ solubility implies more $CH_4$ dissolved in pore water and within bubble streams, potentially increasing the amount of $CH_4$ dissolved in bottom water. A small but significant (R=0.33) positive correlation between $CH_4$ solubility and concentration at site $O_{246}$, and site $O_{91}$ (considering the same time period, i.e. until 3 October in 2015), could indicate such an effect. This is also an alternative explanation for the negative correlation between temperature and $CH_4$ concentration at site $O_{246}$.

### 4.4 Pore water seepage

Short-term temperature increase further offshore (390 m depth) has been linked with release of warm, $CH_4$ rich fluids from the sediments triggered by short duration seismic events (Franek et al., 2017). This means that increased $CH_4$ concentration should be accompanied by increased water temperature and reduced salinity due to admixture of warmer, less saline pore water. We compared short-term anomalies (i.e., deviations from daily means) in these three variables in the 24-hour data sets at both seep sites, but found no corroborating evidence for this hypothesis. Instead, the covariance between current velocity and temperature and salinity anomalies indicates that short-term variability is mainly caused by cross-shelf exchange of Atlantic water in the West Spitsbergen Current and the colder, fresher Arctic water in the Coastal Current due to eddies (Hattermann et al., 2016). It also indicates that $CH_4$ release comes mainly from bubble dissolution and not from pore water seepage.

### 4.5 Seasonal variation of $CH_4$ distribution at site $O_{91}$

Low release of $CH_4$ to the atmosphere from the $O_{91}$ seep area during summer despite high seabed influx, has been explained by suppression of vertical mixing by strong stratification (Myhre et al., 2016a) or absence of mechanical forcing such as wind stress (Silyakova et al., 2020). However, in fall and winter, the water column offshore Prins Karls Forland is expected to have more horizontal and vertical mixing due to weaker stratification from cooling or sea ice formation (Tverberg et al., 2014), baroclinic instability in the frontal structures of the West Spitsbergen Current (von Appen et al., 2016; Hattermann et al., 2016), and more frequent storms (Nilsen et al., 2016).

We expect lower $CH_4$ variability and lower $CH_4$ concentration during periods of high mixing and dispersion, due to weaker

horizontal and vertical gradients and more efficient dispersion of $CH_4$ away from sources. We use three sets of parameters to evaluate long term changes in the amount of mixing in the water column (see Appendix E): i) the 4-week averaged bulk velocity shear ($S_b$), ii) the two dimensional correlation between wind stress and current velocity ($R_{WC}$), and iii) the number of stormy days defined by persistent winds $>11$ ms$^{-1}$ lasting longer than 6 hours (Figure 6). Calm weather, low $S_b$ and $R_{WC}$ until mid-September 2015 indicate a stable water column with limited mixing in the bottom waters. From mid-September, $S_b$

increased and stayed high until mid-November, together with a gradual increase in $R_{WC}$ which can be attributed to a gradual breakdown of stratification and increasing number of storm events (Figure 6a). $R_{WC}$ remained high ($R_{WC} >0.5$ at 60 m depth) until March 2016, indicating a significant effect of wind forcing in the water column. From March until observatory retrieval, $R_{WC}$ decreased to $< 0.2$ below 50 m depth while $S_b$ increased below 60 m depth, indicating available energy for mixing in the bottom waters.

We quantified $CH_4$ variability during the 24-hour measurements using the Median Absolute Deviation (MAD) and used the median as a measure of the amount of dissolved $CH_4$. The three 24-hour periods collected during the calmer period prior to mid-September had high median concentration ($>300$ nmol L$-1$) and the overall highest variability (MAD$>160$ nmol L$-1$), as expected for low mixing conditions (Figures 6b and 6c). From mid-September until the end of March (i.e. fall/winter season), the 24-hour $CH_4$ concentration time-series had generally lower MAD and median concentration. In this period, $CH_4$

variability and median also showed a good statistical relationship with the 5 days accumulated wind stress (R=-0.82 for MAD and R=-0.61 for median concentration), indicating that wind forcing has a deep impact on mixing and redistribution of $CH_4$ in the water column (which also fits well with a high $R_{WC}$). The two last 24-hour $CH_4$ time series (10 April and 1 May) had low median concentration, which could be explained by the absence of stratification (Silyakova et al., 2020) and generation of mixing from the observed increase in $S_b$.

Accumulated wind stress, $S_b$ and $R_{WC}$ are only limited indicators on water column dispersion and mixing. Nonetheless, the relationship between these parameters and the MAD and medians of the 24 hour period $CH_4$ time series gives a good indication on the seasonal cycle of distribution and vertical transport of $CH_4$: strong stratification, less wind forcing and eddy activity in summer limit mixing and prevent $CH_4$ from reaching the atmosphere. However, in fall and winter, reduced stratification makes the water column more prone to mixing and distribution of $CH_4$ seems to be strongly linked with wind forcing from September

to April.

## 5   Conclusions

Time-series of dissolved $CH_4$ at both lander locations show considerably higher $CH_4$ concentrations (up to $1748\pm142$ nmol L$^{-1}$ at $O_{91}$ and $2727\pm182$ nmol L$^{-1}$ at $O_{246}$) than previously found in ship-based water sampling surveys (maximum of 482 near $O_{91}$ and of 564 near $O_{246}$). The time-series also uncover high $CH_4$ variability (up to $\sim1000$ nmol L$^{-1}$) within

335 short timescales ($< 24$ hours), highlighting the uncertainty of flux/inventory estimates based on interpolation/extrapolation techniques where even/linear $CH_4$ distribution is assumed. We calculated the standard error of a mean estimate based on a

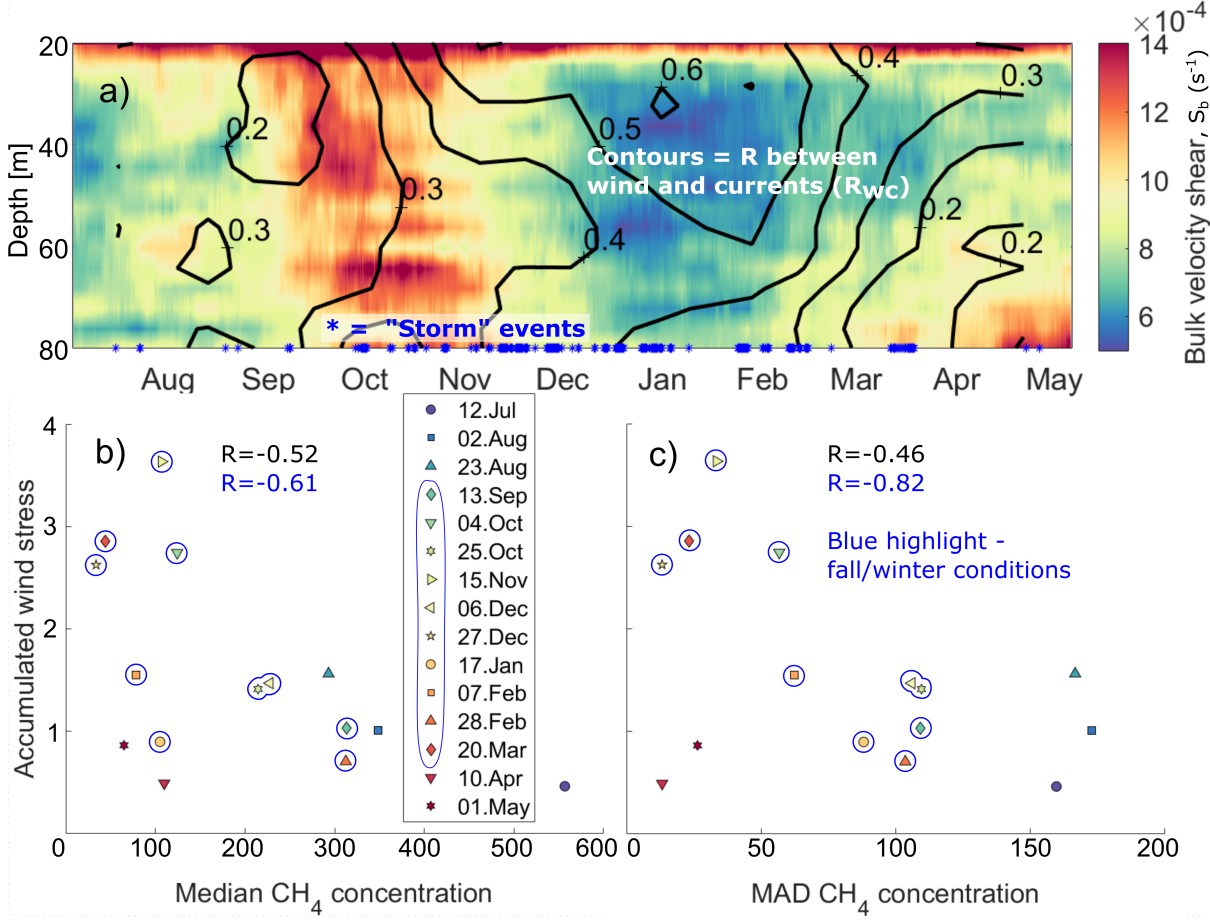

**Figure 6.** a) Bulk velocity shear ($\Delta H = 8$ meter) and two dimensional correlation with wind stress (contours). Relationships between 5 days accumulated wind stress and median (b) as well as median absolute deviation (c) of $CH_4$ concentration for 24 hour data periods. Persistent wind events with more than 10 m s$^{-1}$ winds in periods over 6 hours are indicated with blue stars along the x-axis of diagram a). Blue highlights fall/winter water column conditions as described in the text.

hypothetical discrete water sampling survey based on a range of samples by using the 24-hour time-series as the underlying distribution. The results aligned well with previous discrete water sampling surveys in the area, giving a standard error of the mean of 5-15% for ∼60 samples.

Variability can be linked to directional ocean current variations occurring at tidal time-scales which shows the importance of taking the current direction and seep locations into account when interpreting intense seep site observations. The persistent relationship between current direction and location of seeps during recovery shows that there was seepage throughout the year and that the seep configuration was relatively constant.

  We did not observe a direct effect of tidal pressure variations on $CH_4$ release, but this could be hidden by the strong effect

of variations in current direction. A negative (long-term) correlation between temperature and dissolved $CH_4$ at $O_{246}$ is in agreement with the hypothesized seasonal blocking of lateral $CH_4$ pathways in the sediments (Veloso-Alarcón et al., 2019) but could also be explained by increased $CH_4$ solubility in the water column.

  Short-term, small-scale variations in temperature and salinity were not linked with increased amounts of dissolved $CH_4$, but rather with cross-frontal exchange of water masses due to eddies.

We observed a seasonal cycle in the characteristics of the 24-hour time-series which fits with seasonal changes in dispersion and mixing characteristics of the water column. Higher $CH_4$ concentration and variability in early fall, when stratification was strong, was followed by lower median concentrations and variability in late fall/winter when the water column was more affected by mixing. In late fall/winter, wind forcing was statistically coupled to the concentration and variability of $CH_4$, probably due to weaker water column stratification.

When estimating the atmospheric impact of a particular $CH_4$ source based on sparse measurements, it is crucial to have some constraints on the temporal and spatial variability. These constraints can either be direct knowledge about variability itself or how inventory and fluxes are affected by related physical and/or chemical parameters. We observed considerable temporal and spatial variability at the two seep sites which need to be taken into account to obtain meaningful estimates of $CH_4$ fluxes or inventories. That no strong direct link was found with other oceanographic parameters illustrates the non-linearity of the

system, making careful interpretation of measurements important. Future studies should aim to identify the errors that arise via different up-scaling/interpolation techniques, how these errors can be mitigated, and the methodology optimized. Based on our observations, we suggest that uncertainties in $CH_4$ inventory and seep estimates can be mitigated by taking the local seep configuration, ocean currents and mixing rates into account and employ autonomous instrumentation capable of resolving the steep horizontal gradients in dissolved $CH_4$. This, alongside direct measurements of seepage by e.g., acoustic instrumentation,

can help constrain future estimates of $CH_4$ flux to the atmosphere from seabed seepage.

*Code and data availability.* All data presented in this paper can be obtained upon request to the authors and will also be made available in the platform Open research Data at the University of Tromsø – The Arctic University of Norway (https://dataverse.no/dataverse/uit). All computer code being used can be obtained upon request to the corresponding author

## Appendix A: The K-Lander

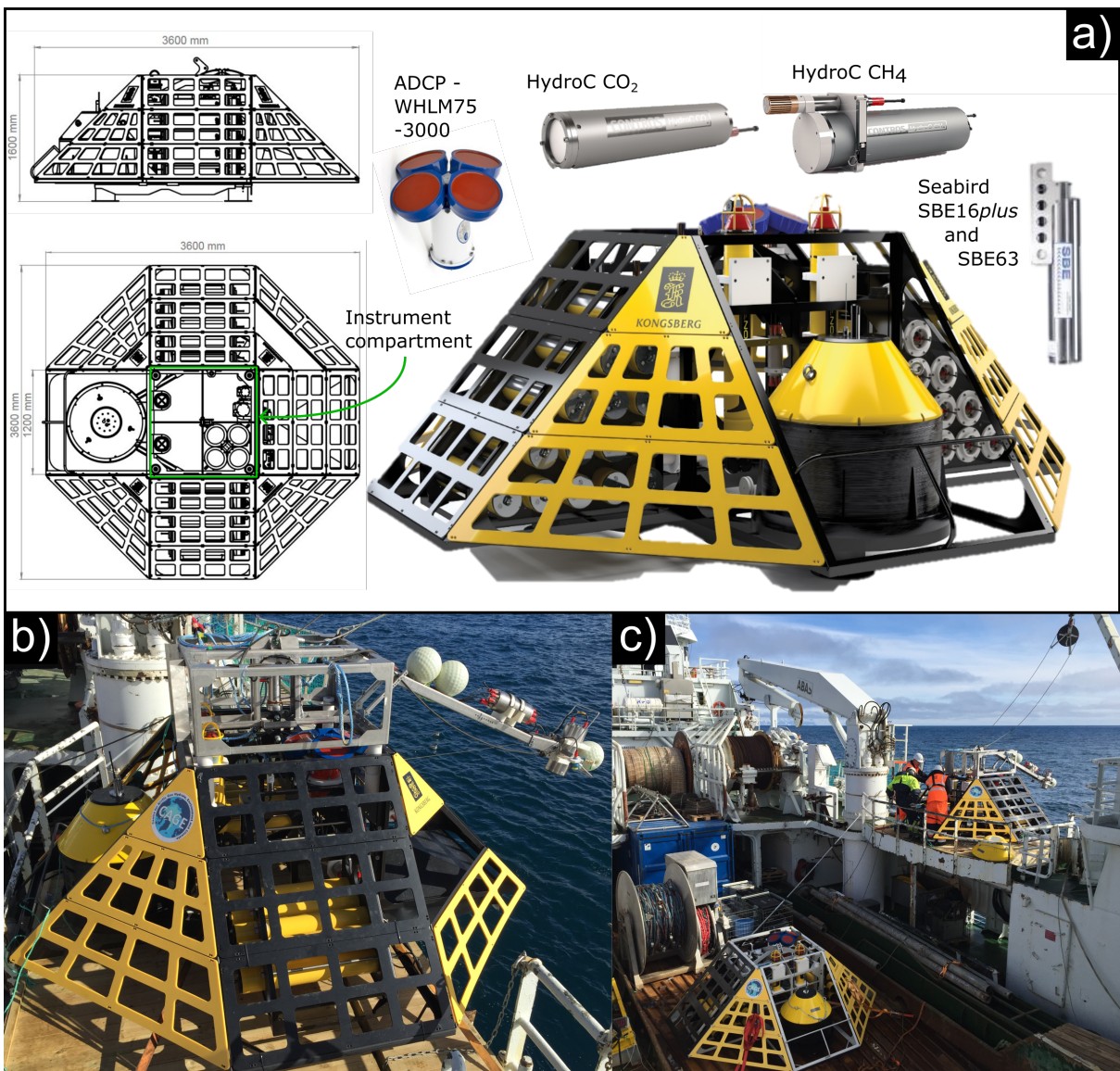

**Figure A1.** a) The K-Lander is a 1.6 m high and 3.6 m wide trawl-proof stainless-steel frame with multiple instrument mounts and batteries. Side panels are perforated to allow unobstructed water flow to the instruments inside the structure. See Appendix B for details on instrumentation. b) K-Lander during deployment with launcher mounted on top and camera system mounted on a boom for visual control of landing area. c) The two K-landers before deployment.

## Appendix B:  Measurement intervals, general post-processing and data

The CTD/oxygen sensor and ADCP conducted measurements every 4 and 9 minutes, respectively, during the continuous monitoring of $CH_4$ and $CO_2$ measurements, and 21 and 29 minutes during the rest of the deployment period (see Table B1 for acronyms, description, and measurement accuracy). Salinity was measured on the practical salinity scale.

The upward mounted ADCP measured ocean currents in 1 m bins with a bottom 7 m blank distance, where the topmost 20% of the water column was disregarded due to side lobe interference. The high resolution, relatively short ensemble time (1 minute), and potential presence of $CH_4$ bubbles in the water resulted in noisy data. We dampened the noise by first removing any data points with error velocities exceeding one short-term (1 week) standard deviation, smoothed the data using a second order Butterworth low-pass filter with a 3-hour cutoff period and a spatial (i.e. vertical) moving average filter with a 5 m Hann window (increasing the blank distance to 10 meter). The accuracy of the ADCP data is therefore not explicitly constrained and is based on comparing current velocity frequency spectra before and after filtering, combined with averaged error velocity of the raw data (Table B1).

**Table B1.** Instruments mounted on $O_{91}$ and $O_{246}$ (see Figure A1), measured parameters, height in meters above sea floor (masf) and stated accuracy. ADCP stands for Acoustic Doppler Current Profiler. N shows the number of data-points used for later multi-variable analysis for $O_{91}/O_{246}$. (*)The Contros HydroC $CH_4$ output partial pressure from the internal gas chamber. (**)We report absolute concentration in seawater (nmol $L^{-1}$) using Henry's law and (***) report accuracy only for response time corrected (RTC) concentration (see Figure B1) since the accuracy for untreated $CH_4$ concentration data is ambiguous due to the slow response time.

| Instrument | Parameter(s) | masf | N | Accuracy |
|---|---|---|---|---|
| Teledyne RDI ADCP WHLM75-3000 | Current velocity Profile | 1.6 | 17438/4731 | $\sim$3 cm s$^{-1}$ |
| Contros HydroC $CH_4$ | $pCH_4$ (instrument output)* $xCH_4$ (reported**) | 1.2 | 1491/281 | $\sim$ 5-20%(RTC***) |
| Contros HydroC $CO_2$ | $pCO_2$ | 1.2 | 1491/281 | N/A (no pump) |
| SeaBird SBE16*plus* V2 | Conductivity/Temperature /Depth | 1.2 | 29660/9065 | 0.0005Sm$^{-1}$/0.005$^o$C, /0.02% of range |
| Seabird SBE63 oxygen optode | Dissolved Oxygen | 1.2 | 29660/9065 | 3$\mu$mol kg$^{-1}$ or $\pm$2% |

Since sensors were recording at different frequencies, chronological alignment of the data was carried out by identifying nearest neighbor data points or by resampling. For correlation coefficients, histograms, and Fourier analysis, the data sets were resampled to a uniform 15 minute or 1 hour measuring interval depending on the sample frequency of the raw data, using a poly-phase anti-aliasing filter. Due to the power-cycling mode of the $CH_4$ and $CO_2$ sensors and differing sampling frequencies, some statistics were based on more data points than others (outlined in Table B1). Daily measurements of $CH_4$ were excluded from these statistics due to the high probability of systematic errors induced by periodic diurnal effects.

Harmonic analysis of hydrostatic pressure and ocean currents was done using t_tide (see Pawlowicz et al., 2002) and the fast Fourier transform.

We calculated the rate of change (ROC) in $CH_4$ concentration using the response time corrected $CH_4$ data and the absolute value of the three point (9 minutes) finite differences to limit the effect of noise on the calculation.

     The absolute concentration of $CH_4$ in the water (nmol $L^{-1}$) was estimated from the partial pressure of $CH_4$, pressure, temperature, and salinity, using Henry's law and Henry constants obtained from Harvey et al., (1996) and practical molar volume and gamma term from Duan & Mao et al., (2006).

The $CH_4$ sensors were calibrated to relevant water temperatures prior to deployment. The TDLAS detectors (Contros GmbH, 2018) provide measurements with good selectivity (fit for purpose), high long-term stability (intermittent calibration not necessary), and are unaffected by dissolved oxygen content (unless complete depletion). Biofouling was also minimal at retrieval (due to the cold water and local setting) and the PDMS membranes are almost unaffected by cold water. Generally, we did no observations indicating issues with any of the sensors except for what already mentioned regarding the conductivity probe

and electrical malfunction of $O_{246}$. Furthermore, we discarded all data recorded during instrument warm-up (i.e. when internal temperature was below correct operating temperature), before the individual measurement periods (the instruments were turned on ~35 minutes prior to recording the data used in the analysis).

     In Contros HydroC $CH_4$ and $CO_2$ sensors, dissolved gases diffuse through a hydrophobic membrane into a gas chamber which equilibrate with the ambient environment. This results in a slow response time (e.g. $\tau_{63}$ ~50 minutes under certain

conditions for our membrane and pump setup for the $CH_4$ sensor) and poor representation of the rapid changes in $CH_4$ we expected in our study area (Gentz et al., (2013) and Myhre et al., (2016)). We therefore performed a response time correction of the dissolved $CH_4$ data following the methodology presented in Dølven et al. (in review, 2021), modulating the response time using the temperature data (effects of salinity on membrane permeability was not taken into account since these are negligible for the local ranges, see Robb (1968)). The $CO_2$ sensors had a faulty pump, which ambiguously increased the response time of

the sensors making response time correction impossible.

     The response time correction was performed for each period individually (1 hour and 24 hour, i.e. 377 periods), using the stated measurement accuracy of the instrument (2 $\mu$Atm or 3% of measured value, whichever is higher) as input uncertainty. We first identified the ideal $\Delta$ t according to the maximum curvature point in the L-curves of the 24 hour measurement periods. These varied slightly between each measurement period, but averaging close to 180 s (176.4 s). To keep the same measuring

interval for all the $CH_4$ data, we therefore corrected all the data with a specified $\Delta$ t of 180 s, which falls well within the bend of the L-curve and should therefore safeguard a good balance between noise and model error (Figure B1a). Inspection of model fit residuals showed a slight modulation following the variance in the signal, explained by our choice to use the same 3-minute measurement grid across a relatively wide variance range, but were otherwise Gaussian. Although expected, this indicates that errors might be slightly overestimated for low-variance sections of the time-series and vice versa for high-variance sections.

The uncertainty estimate varies depending on the amount of $CH_4$ measured by the TDLAS unit in the measurement chamber of the instrument. The distribution of the uncertainty estimates is shown as percentages in Figure B1b. Estimated uncertainty ranged from 3 to 205 nmol $L^{-1}$ (95% confidence, high for high concentrations in measurement chamber and vice versa) or

usually between 5 and 20% although with some outliers when the concentration is low and uncertainty estimate high (Figure B1b).

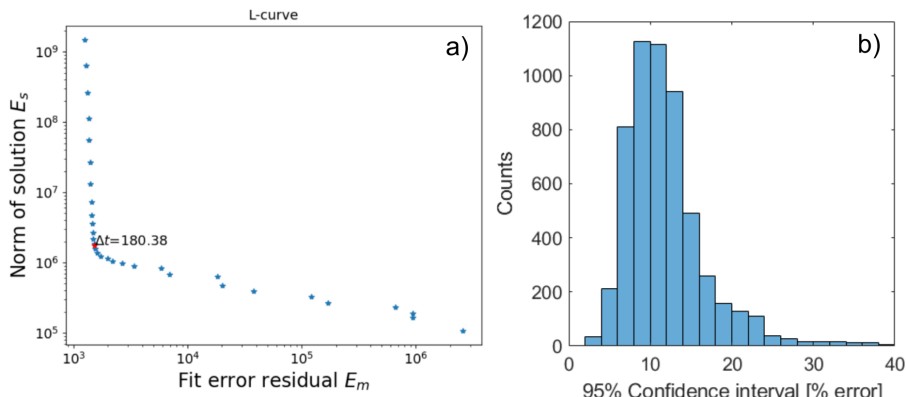

**Figure B1.** a) L-curve for response time correction of $CH_4$ data showing the location of the chosen $\Delta t$ (180 s) for 6 May at O91. b) Estimated relative (percent, %) uncertainty for response time corrected $CH_4$ data (both observatories).

**Appendix C:  24-hour measurements of CH$_4$**

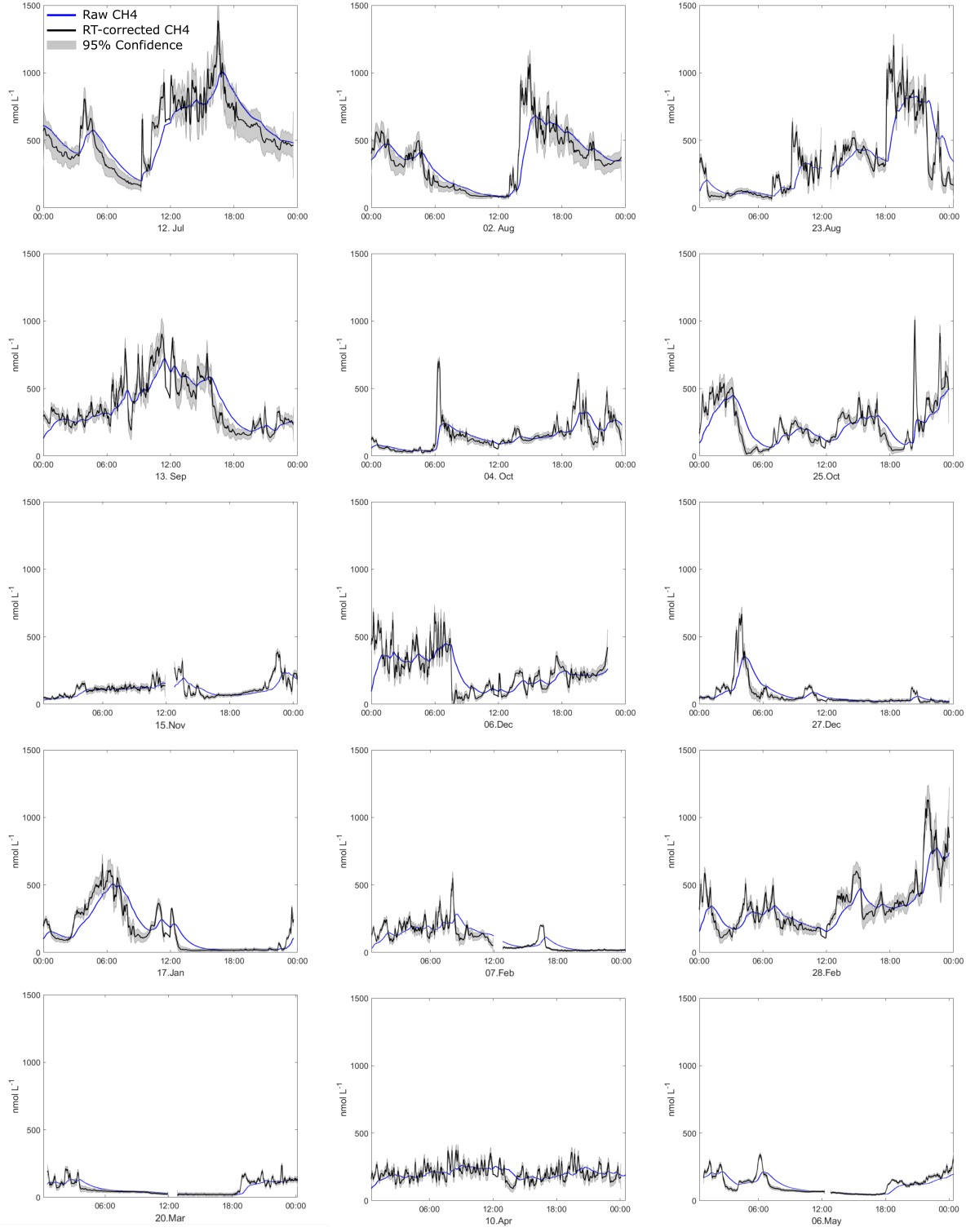

**Figure C1.** All 24 hour periods of $CH_4$ concentration at $O_{91}$ with response time corrected data (black) with uncertainty estimate (grey shade, 95% confidence) and raw data (blue) from $O_{91}$.

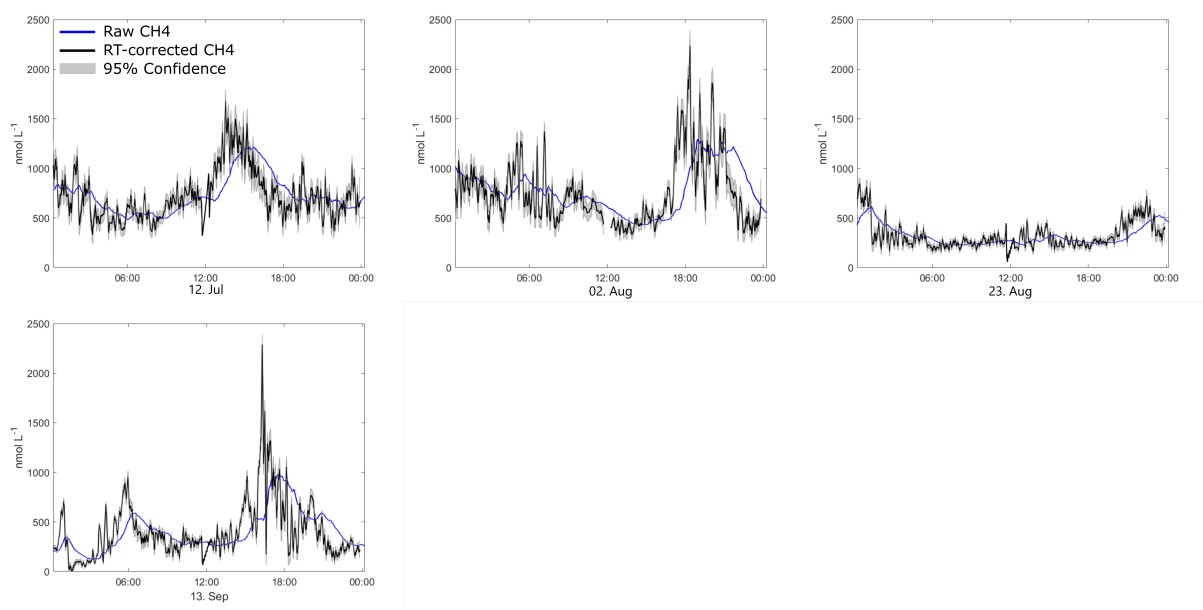

**Figure C2.** All 24 hour periods of $CH_4$ concentration at $O_{246}$ with response time corrected data (black) with uncertainty estimate (grey shade, 95% confidence) and raw data (blue) from $O_{246}$.

## Appendix D: Standard error of mean estimate due to temporal variability

To obtain the (theoretical) true dissolved $CH_4$ average or inventory for an area requires known concentration everywhere at a single point $T_0$ in time. Considering a hypothetical ship-based discrete water sampling survey, any small scale spatial variability not resolved by the sampling grid or localized (not seep-wide) short-term temporal variability occurring during the survey time can be considered measurements errors for the purpose of the survey. Assuming that the water samples are sufficiently spaced out to be considered independent samples, the estimated average concentration from $N$ samples in a particular depth layer in a seep site can be expressed as

$$E(m, \epsilon_t, \epsilon_s) = \frac{\sum_{n=1}^{N}(m + \epsilon_{tn} + \epsilon_{sn})}{N}, \tag{D1}$$

where $m$ is the average of the seep site at $T_0$, $\epsilon_t$ is errors due to temporal short-term deviation from $m$ at sampling time $T_0 + \Delta t$ and $\epsilon_s$ is spatial deviations in concentration from $m$. The expected standard error of $E(m, \epsilon_t, \epsilon_s)$ from the short-term temporal/spatial variability is then given by

$$\sigma_{E(m, \epsilon_t, \epsilon_s)} = \frac{\sigma}{\sqrt{N}} \tag{D2}$$

where $\sigma$ is the standard deviation of the distribution we sample from (Ayyub and McCuen, 2011). From Eq. D1 and Eq. D2 we obtain

$$\sigma_{E(m, \epsilon_t, \epsilon_s)} = \sigma_{E(m, \epsilon_t)} + \sigma_{E(m, \epsilon_s)} = \frac{\sigma_t}{\sqrt{N}} + \frac{\sigma_s}{\sqrt{N}} \tag{D3}$$

where $\sigma_t$ and $\sigma_s$ is the $\epsilon_t$ (temporal), and $\epsilon_s$ (spatial) variability related standard deviations of the distribution and $\sigma_{E(m, \epsilon_t)}$ and $\sigma_{E(m, \epsilon_s)}$ the corresponding contributions to the standard error of the mean. Assuming the daily variance at the observatory is representative for the seep site, we can describe the expected error caused by sub-daily variability (all $\epsilon_t$) in a scenario where a seep site is being sampled $N$ times using the 24-hour time-series as the underlying distribution. In essence, we treat every measurement as having an associated probability distribution which is represented by the 24-hour time-series (which gives the sub-daily variability).

In the discrete water sample data presented in Silyakova et al. (2020), the underlying distribution is unknown and we can only assume that the sample distribution resembles the underlying distribution, i.e. that

$$\sigma_{E(m, \epsilon_t, \epsilon_s)} \approx \hat{\sigma}_{E(m, \epsilon_t, \epsilon_s)} = \frac{\sigma_{sampled}}{\sqrt{N}}, \tag{D4}$$

where $\hat{\sigma}_{E(m, \epsilon_t, \epsilon_s)}$ is the standard error estimate of the mean based on the sample distribution and $\sigma_{sampled}$ is the standard deviation of the measurements. All three data sets, "June-14" ($N$=64), "July-15" ($N$=62), and "May-16" ($N$=63), have similarly

skewed distribution compared to what is found in the observatory data (see Figure D1), which supports this assumption. The survey in Silyakova et al. (2020) required 2-3 days to complete, while the observatory data only concerns sub-daily variability (24-hour time-series). Nonetheless, we believe the comparison is valid, since the known major contributors to short-term (time-scales below weeks) variability acts on sub-daily (or at least $\leq$ daily) scales, such as the dominant frequencies in the ocean currents and pressure changes.

There is a clear relationship of increasing $\sigma_{E(m,\epsilon_t,\epsilon_s)}$ with increasing daily average, making relative $\sigma_{E(m,\epsilon_t,\epsilon_s)}$ a meaningful quantity to use, as opposed to absolute $\sigma_{E(m,\epsilon_t,\epsilon_s)}$. Additionally, for simplicity, we have not differentiated in the notation of the standard error of the mean (SEM) in the main text of the manuscript, referring to it as simply SEM in all situations.

It is also enlightening to consider the distribution of average estimates and how the skewed underlying distribution affects the distribution of average estimate errors for smaller $N$. We did this by simulating hypothetical surveys by random sampling from the 24-hour data-sets (Figure D2) which shows the elevated probability of underestimating the average for estimates based on few samples ($N \lesssim 30$), i.e. the median error is smaller than the average error. This is caused by an inheritance of the skewed underlying distribution in the $CH_4$ concentration data (see Figure D2a). This also allows for severe overestimates due to the long right-hand side tail of the distribution. For larger $N$s ($N \gtrsim 30$), average estimates tend towards being normally distributed, thus avoiding these effects (see Figure D2b).

Error estimates of more complicated properties, such as the total $CH_4$ content in a volume of water based on interpolation techniques, require an assessment of the individual uncertainties of each measurement and how these errors propagate via e.g. linear interpolation in the spatial domain. While not being explicitly applicable to inventory estimates, the $\sigma_E$ still describes how random errors cancel out for larger $N$s in evenly sampled grids, assuming this variability is representative for the seep site.

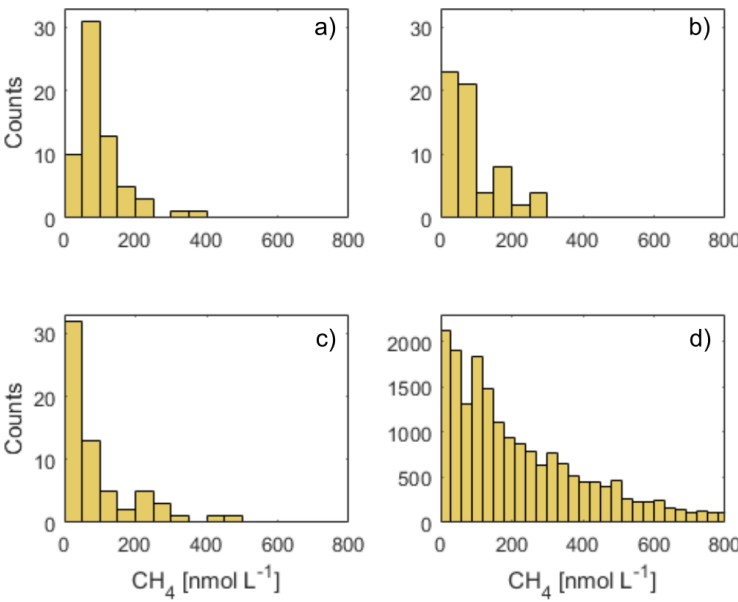

**Figure D1.** Distribution of $CH_4$ concentrations from the a) June-14 b) July-15 c) May-16 data in Silyakova et al. (2020) and d) from the 24-hour data (all periods) at $O_{91}$. Note the different scale for the y-axis between a-c and d.

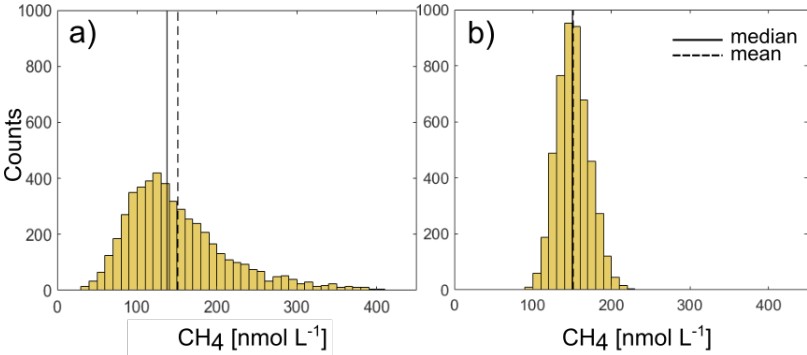

**Figure D2.** Histograms of simulated average estimates based on $N=10$ (a) and $N=30$ (b) samples from the 24-hour data set from 23 August at $O_{91}$ showing the median and mean as vertical lines.

## Appendix E:  Bulk velocity shear and wind stress correlation

We calculated bulk wind stress using 10 meter above sea level ERA-interim re-analysis wind data (Dee et al., 2011) and Large & Pond (1981). Water column bulk velocity shear $S_b$ (see e.g. Lincoln et al., 2016) was calculated as

$$S_b^2 = \left( \frac{u_u - u_l}{h_{diff}} \right)^2 + \left( \frac{v_u - v_l}{h_{diff}} \right)^2 \tag{E1}$$

where $u_u$, $u_l$, $v_u$, $v_l$ refer to the easterly and northerly ADCP velocity components in the upper (subscript u) and lower (subscript l) layer and $h_{diff}$ the vertical distance between layers. The direct effect of wind stress is usually confined to surface water, although indirect effects such as Ekman transport/overturning and the formation of eddies can facilitate currents and mixing at deeper depths (Cushman-Roisin and Beckers, 2011). The two-dimensional correlation coefficient $R_{WC}$ between the wind and ocean currents was calculated using Kundu, (1976) and the complex representations $\tau_c$ and $u_c$ of the wind stress and de-tided current velocity vectors:

$$R_{WC} = \frac{\langle \tau_c^* u_c \rangle}{\langle \tau_c^* \tau_c \rangle^{\frac{1}{2}} \langle u_c^* u_c \rangle^{\frac{1}{2}}} \tag{E2}$$

where $\langle .. \rangle$ gives the normalized inner product of the vectors and $^*$ annotates the complex conjugate. We allow time-lags up to 15 hours to account for the gradual and indirect effects of wind stress on the ocean currents. Both properties were estimated throughout the valid current velocity profile, but only down to 80 m depth due to the 8 m vertical distance between the defined layers used in the bulk velocity shear calculation.

*Author contributions.* Conceptualization: KOD,BF,AS,PL,PJ. Data curation: KOD,BF,MM. Formal Analysis: KOD,BF,MM. Funding acquisition: BF. Investigation: KOD, BF, AS. Methodology: KOD,BF. Project administration: BF, AS. Resources: BF. Software: KOD. Supervision: BF. Validation: n/a. Visualization: KOD,MM. Writing - original draft preparation: KOD. Writing - review & editing: KOD, BF, AS, PL, PJ, MM.

*Competing interests.* The authors declare that they have no conflict of interest.

*Acknowledgements.* We thank the crew of R/V Helmer Hanssen during the deployment (CAGE 15-3) and recovery (CAGE 16-4) cruises. This study is a part of CAGE (Centre for Arctic Gas Hydrate, Environment and Climate), Norwegian Research Council grant no. 223259. We thank Nicholas Warner for proofreading the article.

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
