# Peer review of "Autonomous methane seep site monitoring offshore Western Svalbard: Hourly to seasonal variability and associated oceanographic parameters"

_Ocean Science, 2021_

## Author Response (AR1)

**Author's response**

The following replies are mainly directly based on (i.e. cited from) the interactive discussion ("AC1", "AC2", and "AC3"). Replies reproduced here are truncated where we saw fit and line numbers updated to fit the uploaded revised manuscript (the non marked-up version). In other words; OBS: Line numbers to edits in AC1 and AC2 are not correct after final edits, but they are correct in this document. Single words have been changed from AC1 and AC2 at a couple of places where I found sentences to be unclear. Any substantial additions/changes to AC1, AC2, or AC3 is colored purple and marked with "Addition to (…)". Reproduced referee comments are not edited in any way. The replies are structured to comply with the suggested structure of the "Author's response" with (1) referee comment (2) author comment (3) implemented revisions. Referee comments are in blue-italic and author response in black font. Implemented changes are highlighted in red font. I have emboldened certain parts of the text in RC1 to make it easier to compare and see the structure in a point by point fashion. All figures referred to in the responses have been uploaded in a "supplement to author's response" pdf file.

*RC1: 'Comment on os-2021-85', Anonymous Referee #1*

*The manuscript by Knut Ola Dølven and colleagues is a very interesting in-situ study of methane seeps west of Spitsbergen. Its conclusions about variability of the dissolved gas concentrations are very important for planning any future measurement campaigns. The information on possible increase in methane seepage is interesting and its (cautious) discussion is correct, in my opinion. In short this study definitely publishable and should be published after only some minor corrections.*

*The few things I would like the authors to address are:*

1) *The salinity used in this study is obviously practical salinity (no unit), not absolute salinity (g kg$^{-1}$). This fact (the name "practical salinity") should be mentioned somewhere (something like "here and elsewhere in the papers the salinity values are practical salinity").*

Line 98: *"here and elsewhere in the paper, salinity values are practical salinity"*

2) *Are the CH4 and CO2 trends described in lines 110 and 115 statistically significant, considering the high variability of the measurement values? The former (methane) probably is due to the large difference at the start and end of the time series, but I would not be sure about the latter (CO$_2$). Also, generally no tests of statistical significance or*

In these lines we describe a 2-week median value including both 24-hour and daily 1-hour data, which prevents from calculating meaningful statistical significance due to persistence and an uneven sampling scheme. However, the trend described in the time period (up until January 2016) is statistically significant, with a p-value <0.001 for an F-test with N-2 degrees of freedom (see e.g. James et al., 2013) and a Durbin-Watson statistic of 1.78 indicating no autocorrelation (upper limit, dU=1.72, see Durbin and Watson, 1971), using the averaged 1-hour data, including the first hour of the 24-hour weekly data for a total of 190 days/datapoints from July to January, assuming a linear trend. A caveat with relying on the daily 1-hour data in this way is the 23.93 hour periodicity in the current velocity data, which gradually changes the most

probable origin of the measured water. This effect is described in Sect. 4.1 ln 176 in the revised manuscript.  Using only averaged 24-hour data (avoiding the

problem of daily periodicity in current direction), i.e. 1 value every 21 day to a total of 9 values (up to 28 December), gives a non-significant trend (p=0.09). It should, however, also be noted that we have assumed a linear trend, which is probably not the best regression model in this case.

Due to the faulty pump and inherent uncertain response time for the $CO_2$ sensor, we could not use the daily 1-hour data. Therefore, the only (if any) meaningful way to report significance for this data would be to average the data collected in the 24-hour measurements. In that case, the trend from July to November is not significant (p=0.39), but the decreasing trend from November to May is (p=0.0001). But as described in the manuscript, this data should be interpreted with caution due to the faulty pump.

In general, we decided not to focus on reporting statistical significance for the different trends and correlations due to the uneven sampling scheme, wide range of characteristic time-scales in the chemical/physical mechanisms we investigated, nonlinearity, as well as persistence at different time-scales in the different datasets. For instance, temperature autocorrelates on time-scales of weeks, while methane concentration autocorrelates on time-scales less than a few hours. Indeed, the correlation matrices in the Results section are meant as a first order overview of potential relationships. While there are ways to circumvent problems with persistence in time-series, e.g. by using autoregressive models or estimating the effective degrees of freedom (Bretherton et al., 1999), we would in any case have to individually adapt methodology for each time-series to obtain meaningful significance estimates. We therefore instead chose to focus on identifying the principal features, phenomenon, and hypothesis that should be further investigated and provide individual, tailored analysis of these (the Discussion section).

3) *A minor nitpick. The statement in line 115 can't be right as it is written: "$CO_2$ averaged 403 µatm with a decrease from mid-November 2015 (~400 µatm) until 6 May (~391 µatm) in 2016 (Figure 2a)". Looking at the figure I know what you mean but still something that decreases from 400 to 391 should not have on average 403. Please rephrase.*

This is now clarified in line 121

Line 121: "$CO_2$ averaged 403  µatm with an increase towards mid-November 2015 (410 µatm) then a decrease until 6 May (391 µatm) in 2016 (Figure 2a)."

4) *The cited literature is very rich and generally well chosen but I am surprised by the lack of a citation of the review paper James et al. 2016, https://doi.org/10.1002/lno.10307. It could be cited in multiple places in the manuscript as it covers many of its threads. For full disclosure I am one of its co-authors so please treat this as non-obligatory but I honestly think it's lack is puzzling.*

James et al., 2016 is now appropriately cited in line 33 and 36.

*RC1: 'Comment on os-2021-85', Anonymous Referee #1*

*Dølven et al. present two very interesting time-series of bottom water measurements of physical and chemical parameters from two autonomous in-situ ocean observatories at methane seeps west of Spitsbergen. The study shows high short- and longer-term variations, as well as higher methane concentrations compared to previous studies. Discussion and conclusions are refreshingly kept cautious and focusses on the temporal variability which might be caused by various factors.*

*The study is generally well written and sections are clearly structured. However, I do have some major concerns about the quality of the methane data. By this, I recommend this study for publication after major revisions based on the comments, I listed below:*

***Methane sensor***

*Measuring methane and other gases in the water column is very challenging and prone to errors, especially when sensors are deployed over a longer period. The authors base the results of their study on $CH_4$ concentration data obtained by two HydroC $CH_4$ sensors. These "simple" sensors are known to have strong limitations for providing quantitative measurement, due to its **low selectivity** and **strong dependency to changes in the physical conditions** (e.g. **biofouling, hydrostatic pressure, water temperature, salinity and dissolved oxygen content). How did the authors manage to calibrate and validate their measurements, e.g. by discrete sampling and subsequent laboratory analysis before, during and after deployment?** The material & methods section of this study is rather weak concerning these measurements. **I miss information about the accuracy, precision, resolution, and sensitivity of the deployed sensors.** How do the sensors (especially for methane) behave during the **power-on-off-cycles**, so during the measurements for an hour every day, concerning the reproducibility? Here, the measurement accuracies and precision certainly deviate from a 24h measurement cycle. These deviations should also have a significant impact on the data and the calculated correlation coefficients.*

*In addition**, I wonder if the choice of sensors might result in the large offset from the other studies,** as Gentz et al. used a calibrated (by discrete samples) in situ underwater mass spectrometer and Silyakova et al. based their study on discrete sampling of the water column (**btw: correct the doi for this study**).*

First we would like to point out a couple of general changes we made regarding this topic (also partly based on the review comment). Firstly, we have now clarified what detector, membrane, and pump system we used

Line 87-91: *"The $CH_4$ sensors rely on a Tunable Diode Laser Absorption Spectrometry (TDLAS) detector, while the $CO_2$ sensors use Non-dispersive infrared (NDIR) detectors. Both sensors were equipped with polydimethylsiloxate (PDMS) membranes, and a Seabird SBE 5M pump (see Appendix B)"*

Additionally, since Dølven et al. (2021) is now openly available to the public, we found it appropriate to shorten the description of the response time correction method and merge Appendix C into Appendix B.

Structural change: Appendix C is merged with Appendix B

Appendix B now contains all instrument/measurement related information not crucial to understand the content of the manuscript. We address all concerns mentioned by the reviewer in the reply/explanations below and have implemented corresponding changes/additions in the manuscript.

- **Low selectivity**: This might have been an issue in previous versions of the sensor relying on NDIR, but Tunable Diode Laser Absorption Spectroscopy (TDLAS) detectors such as the one used in the Contros HydroC $CH_4$ have relatively good selectivity (see Figure R1 and e.g. Shemshad et al., 2012).

  Line 389-390: Added information about selectivity.

- The **Accuracy** of the data is described in Appendix B. We used the ISO 5725-1 definition of accuracy which includes both random and systematic errors (sometimes referred to as **precision** (random errors) and *accuracy* (systematic errors)). Appendix B describes both instrument accuracy and accuracy of the response time corrected data, which we use in the results and discussions of the manuscript (Table B1). We agree that these uncertainties could have been more elaborately described, and have therefore added more detailed information on this in line 410-414 in the revised manuscript and a figure showing the distribution of expected errors in the response time corrected data (Figure B1b, see also Dølven et al., 2021). We have also added an explicit address of the uncertainty in the methods part of the main text of the manuscript (line 105-108). Additionally, we added the 95% uncertainty range for all discrete (i.e. not averages) concentrations mentioned in the text.

  *Addition to RC2:* temporal **Resolution** *is explicitly described in Appendix B (3 minutes).*

  Line 410-414: *"The uncertainty estimate varies depending on the amount of $CH_4$ measured by the TDLAS unit in the measurement chamber of the instrument. The distribution of the uncertainty estimates is shown as percentages in Figure B1b. Estimated uncertainty ranged from 3 to 205 nmol $L^{-1}$ (95% confidence, high for high concentrations in measurement chamber and vice versa) or usually between 5 and 20% although with some outliers when the concentration is low and uncertainty estimate high (Figure B1b)."*

  Line 105-108: *"Uncertainty ranges for the $CH_4$ data are reported as 95% confidence intervals and typically vary between 5 and 20% (full distribution of uncertainties are in Figure B1b)."*

  Appendix B: Added Figure B1b showing distribution of expected errors in $CH_4$ of the response time corrected data.

- **Strong dependency to changes in the physical conditions**
  - **Biofouling**: We observed little to no biofouling on the instruments and observatories upon recovery (probably due to cold water and local environment). Figure R2 shows the pumps from the HydroC $CH_4$ and $CO_2$ directly after recovery, i.e. after 10 months in the water. We added a sentence stating that there was only minimal biofouling and no other indications of problems with the sensors other than the electrical

malfunction at $O_{246}$ and the conductivity sensor (all sensors, not only the HydroCs) at retrieval in line 390-393.

Line 390-393: *"Biofouling was also minimal at retrieval (due to the cold water and local setting)"*

- o **Hydrostatic pressure**: The change in hydrostatic pressure during the deployment was small, i.e. between 1.2 and 1.5 dbar over the course of one tidal cycle (~12 hours). The pressure fluctuations in the measuring chamber were also small (R<0.05 dbar) and had a statistically negligible relationship with concentration (which could also be related to other environmental processes, cf. Figure R3).

- o **Water temperature**: The bottom water temperature varied with less than ~3 °C and the internal temperature was kept constant at correct instrument operating temperature for data recorded and used in analysis (we discarded measurements obtained during instrument warm-up). Water temperature indeed affects the response time of the instrument due to changes in membrane permeability, we now explicitly address the effect of temperature changes and how we accounted for this in the response time correction procedure in line 400-401 (see also Dølven et al., 2021).

Line 400-401: *"(…) following the methodology presented in Dølven et al. (2021), modulating the response time using the temperature data (…)"*

- o **Salinity**: The relatively small changes in salinity observed at the measurement sites should only have a negligible effect on membrane permeability (Robb, 1968). This is now also mentioned in Appendix B (line 401-402).

Line 401-402: *"(…)(effects of salinity on membrane permeability was not taken into account since these are negligible for the local ranges, see Robb (1968)(…)"*

- o **Dissolved oxygen content**: This should have no effect on the measurements unless there is a complete depletion of oxygen which is not the case at the observatory sites (Figure 2 in manuscript). Dissolved oxygen can influence sensors relying on metal dioxide detectors (Boulart et al., 2010), but should not affect the TDLAS used in the HydroC $CH_4$. We added this information in line 387-388.

Line 390: *"(…)and are unaffected by dissolved oxygen content (unless complete depletion)(…)"*

- • **Long-term drift/calibration**: Standard calibration procedures (for relevant conditions) were followed prior to deployment. While long-term stability might be an issue with NDIR detectors, post/intermittent calibration was neither recommended by the manufacturer nor found necessary (the latter also practically very difficult) due to the high stability of the TDLAS unit and PDMS membranes which are almost unaffected by cold water (this is also the case for teflon membranes). We added this information in

line 388-390. This is supported by 4h-Jena (and previously Contros) who have aggregated and cross-checked data from sensors that have been returned after long-term deployments over several years and found that any drift or changes are insignificant (<1 ppm both for low and high concentrations) (pers. comm. Jack Triest, 4h-Jena GmbH).

Line 388-390: *"The TDLAS detectors (Contros GmbH, 2016) provide measurements with good selectivity (fit for purpose), high long-term stability (intermittent calibration not necessary)"*

Line 391: *"(…)the PDMS membranes are almost unaffected by cold water. "*

- **The power-on-off-cycles**: As previously mentioned, all measurements obtained during the instrument warm-up period were discarded; in practice, the instrument was turned on approximately 35 minutes before used data was recorded (the 1- and 24-hour periods therefore vary slightly by 1-2 minutes in length). The sensor operation should therefore not be affected by the length of the measuring periods. We added this information in line 393-395.

  Line 393-395: *"Furthermore, we discarded all data recorded during instrument warm-up (i.e. when internal temperature was below correct operating temperature), before the individual measurement periods (the instruments were turned on 35 minutes prior to recording the data used in the analysis)."*

Addition to RC2: The paragraph below from AC2 justifies why **sensitivity** is not very important for our study (I must admit I forgot to explicitly address this in AC2). To address this explicitly we should have mentioned in AC2 that while TDLAS detectors can provide good sensitivity, the response time correction involves some noise amplification to extract the fast response signal (this is reflected in the reported accuracy), which means that the data we present is not very well suited to, nor aimed at describing low concentrations with very high accuracy. Nonetheless, we are not concerned with very low concentrations in our study and very seldom measure concentrations below 16 nmol $L^{-1}$ (the 2.5$^{th}$ percentile at $O_{91}$ and $O_{246}$ is 16 and 107 nmol $L^{-1}$, respectively) and the lowest concentration we measure is 5 nmol $L^{-1}$, which is still considerably higher than the detection limit of the sensor.

Even though modern HydroC $CH_4$s with our pump/membrane/detector setup can give decent accuracy and be applicable in a wide range of settings, it might still be relevant to acknowledge that the analysis and discussions in the current manuscript mainly concerns large changes and high concentrations. Considering this, we believe the response time corrected Contros HydroC $CH_4$ data should be more than sufficient to support the scientific results and inferences described in our manuscript. Additionally, the data show that the sensor can produce high variability, high concentration, and low concentration data throughout the time-series and also produce a stable minimum (background) concentration at around 10 nmol $L^{-1}$ (Figures 2, 4, 6, and Appendix B). Based on this and what iterated above, we found no apparent reason to question the reliability of the sensor for the purpose of the study and believe the additions in the method section and in Appendix B, as well as addition of uncertainty ranges for concentrations mentioned in the results/discussion sections should be sufficient to address this.

Reply on AC2 from authors (AC3) (https://doi.org/10.5194/os-2021-85-AC3):

There is a typo in AC2 where it is stated that there is a stable minimum at around 10 nmol $L^{-1}$. The statement should instead read "minimum at around 12-13 nmol $L^{-1}$". Additionally, it should be clarified that this is for response time corrected data at $O_{91}$. It should also be noted that the concentration do occasionally (although very seldom) decrease below this value. See figure attached to the reply which contains a histogram of the response time corrected 24-hour time-series data at $O_{91}$ with a plotted green vertical dashed line at 12 nmol $L^{-1}$.

***Discrepancy with results in Gentz et al. (2014)*:** We trust that the text reflects that we emphasize high spatiotemporal variability and sparse sampling as a possible explanation for the discrepancy in concentrations when comparing with data from Gentz et al. (2014). Taking the spatiotemporal variability into account, and the fact that $O_{246}$ was deployed around 30 m from an intense seep, we believe the measurements aligns reasonably well with Gentz et al., (2014). It should also be noted that the data in Gentz et al. (2014) was obtained with discrete water samples (not the underwater mass spectrometer, which was only used a 10 m water depth), greatly limiting data coverage compared to a continuous measurement. We also find that the comparison with the concentrations reported in Silyakova et al. (2020) (**the doi should be correct**) seems reasonable taking the above perspectives into account and the similarities in distribution of values (see added Figures showing distributions in Appendix D).

**Methane inventory**

*In the introduction, the authors emphasize the urgent need of continuous measurements to detect the temporal variability and by this validate or correct the $CH_4$ budget of seabed seepages (with which I strongly agree). And in addition, the authors state "We highlight uncertainties in methane inventory estimates based on discrete water sampling" whereas the discussion part only focus on comparing minimum and maximum $CH_4$ concentrations and remains extremely unspecific with "…$CH_4$ concentration at our locations can change by up to 2 orders of magnitude within hours…". Their study, however, lacks any statement and calculation on how their findings will impact these budgets. Here, I miss at least some basic calculations of $CH_4$ inventories and e.g. its variation over time. Furthermore, I would recommend to add a figure (or sub-figure) showing the mins, max and means (or medians) of this study along with results from previous studies (e.g. simple box-plot).*

We agree that more details on the implications for inventory/budget estimates would improve the manuscript. We address this by using a statistical approach where we find the expected error from unresolved short-term variability for a hypothetical discrete water sampling survey seeking to estimate seep site averages, where the short-term variability is represented by the 24-hour time-series of the observatories. This exercise also explicitly describes the expected errors for single measurements. We compare with the results presented in Silyakova et al. (2020) who performed discrete water sampling surveys in the $O_{91}$ area every summer from 2014-2016. The content is added via a remodulation and extension of the last part of section 4.1 ("$CH_4$ variability", lines 204 to 242), Appendix D (lines 414-460), Figure 4, D1 and D2. We concluded that a budget estimate would provide unreliable results since we are monitoring $CH_4$ only at a single location.  We believe the added result strengthens the manuscript and at the same time addresses the reviewers concern about being too unspecific on implications for budget estimates. This addition also led to a slight reformulation and addition in the abstract, introduction, and conclusion of the manuscript (i.e. adding one sentence describing the results).

Line 204-242: Remodulated and added content on quantification of potential errors in ship-based surveys taking the observed short-term variability into account

Added figure in Sect. 4.1: Figure 4.

Line 416-462: Added Appendix D which describes underlying methodology for additions in Line 204-242. Including figures D1 and D2.

Line 353-354: *"Future studies should aim to identify the errors that arise via different up-scaling/interpolation techniques, how these errors can be mitigated, and the methodology optimized."*

Changed last sentence in abstract to: *"We present new information about short- and long-term methane variability and provide a preliminary constraint on the uncertainties that arise in methane inventory estimates from this variability."*

We appreciate the suggestion to add a boxplot and agree that this could illustrate certain aspects of the data in an elegant way. However, in addition to the above reasoning on not including inventory estimates based on our data (which such a boxplot might have shown), we believe that the addition of a boxplot would not add any new information to what is already shown in Figures 2,3,4,6, and Appendix C, which all concern dissolved $CH_4$ data.

**Figure2**

*Figure 2 is the main figure of the manuscript and needs urgent revision: 1) This figure contains all obtained data and should make use of the entire page. Please adjust the height, which will give the reader the chance to recognize some details as well. 2) Data from August 3$^{rd}$ are missing in Figure 2d. 3) Use the same axis scales for O91 and O246 or make the reader aware of different scales. 4) Note the problems with the sensors at O246 in the annotations of the figure. 5) Use identical font and font properties (e.g. boldness) over the entire figure.*

The figure is now extended to the limits of the template, fonts are homogenized (to helvetica) and the different axis and issue at $O_{246}$ are pointed out in the figure caption. Figure 2d shows a continuous 24-hour measurement (obtained every 21 days) followed by 1-hour measurement (obtained every day) on 2-3 Aug. Figure 2d is meant to show both a 24-hour and a 1-hour time-series in the same figure for clarity and has no missing data.

Edits on Figure 2: The vertical dimension of Figure 2 have been increased, text homogenized, axis and issues at $O_{246}$ now specified in figure caption.

Regarding the marked-up manuscript version with tracked changes:

- Latexdiff did not remove the headline of the "old" Appendix C (which is now merged with Appendix B), i.e. "Appendix C" is not stroked out. (The appendix numbering is still correct)
- Latexdiff does not track additions/removals in the reference list.

Other edits

Edits to figure A1 and A2: Figure A2 is now merged into figure A1

Line 111: Changed value for maximum concentration to 1748 (typo). Applied same change when the number reappears in the discussion.

Working with RC2, I realized that it might be appropriate to supply the 95th inter-percentile for the methane data, since both minimum and maximum values are quite "lonely" datapoints. I also found that it might be useful to provide the interquartile range alongside the mean and median values to indicate spread (we are not reporting standard deviations due to the distribution properties, and also not explicitly showing the distribution). These inter-percentiles and inter-quartiles were added in line 111-114 and line 145-147.

Other than this, there are small corrections of typos and unclear formulations here and there throughout the text that were identified during the new readthroughs. These are all marked up in the marked-up version of the manuscript.

**References**

Bretherton, C. S., Widmann, M., Dymnikov, V., Wallace, J. M., Bladé, I.: The effective number of spatial degrees of freedom of a time-varying field. Journal of Climate, 12(7):1990-2009., https://doi.org/10.1175/1520-0442(1999)012<1990:TENOSD>2.0.CO;2, 1999.

Durbin, J., Watson, G. S.: Testing for Serial Correlation in Least Squares Regression. III. Biometrica, 58(1), 1–19. https://doi.org/10.2307/2334313, 1971.

James, G., Witten, D., Hastie T., Tibshirani R.: An Introduction to Statistical Learning: with Applications in R. New York: Springer, 2013.

Shemshad, J., Aminossadati, S. M., Kizil, M. S.,: A review of developments in near infrared methane detection based on tunable diode laser, *Sensors and Actuators B: Chemical*, 171-172, 77-92, 2012. *https://doi.org/10.1016/j.snb.2012.06.018*

Boulart, C., Connelly, D. and Mowlem, M.: Sensors and technologies for in situ dissolved methane measurements and their evaluation using technology readiness levels. *Trends in Analytical Chemistry*, 29(2), 186-195, 2010.

Dølven, K. O., Vierinen, J., Grilli, R., Triest, J., and Ferré, B.: Response Time Correction of Slow Response Sensor Data by Deconvolution of the Growth Law Equation. *Geoscientific Instrumentation, Methods, and Data Systems Discussion*, 1-22, https://doi.org/10.5194/gi-2021-28, 2021.

Robb, W. L.: Thin silicone membranes – Their permeation properties and some applications, Annals of the New York Academy of Sciences, 146, 119-137, https://doi.org/https://doi.org/10.1111/j.1749-6632.1968.tb20277.x, 1968.

[Figure]

**CONTROS HydroC™ CH$_4$**

**Selectivity**

[Figure]

*Figure R1: Selectivity of the TDLAS detector from lab test at 4h-Jena engineering*

[Figure]

*Figure R2: Filter and pump intake for the two CONTROS sensors from O246 directly after retrieval.*

[Figure]

*Figure R3: Pressure in the measuring chamber of the two sensors, i.e. O91 (left) and O246 (right). Only one datapoint is used for each measurement period to avoid effects of autocorrelation (due to the uneven sampling scheme).*

[Figure]

*Figure accompanying AC3 showing the distribution of methane concentrations and a vertical dashed line at 12 nmol/L*

---

## Author Response (AR2)

The replies below are structured to comply with the suggested structure of the "Author's response" with (1) referee comment (2) author comment (3) implemented revisions. Referee/editor's comments are in blue-italic and authors' response in black font. Implemented changes are highlighted in red font.

**Response to reviewer #2.**

The authors thank again reviewer #2 for providing new suggestions and for being rigorous with this review - this is highly appreciated and helped us improve the manuscript.

*I experienced various HydroC sensors that not only show large discrepancies from established methods (tens of nanomoles per liter) but also randomly distributed time series patterns during field deployments in the past, so it's good to read that the new TDLAS-based sensors seem to be more reliable. Hopefully, the statement "post/intermittent calibration was neither recommended by the manufacturer nor found necessary" is not entirely serious... Verification of scientific data is more than essential, even if a manufacturer states that continuous verification and/or calibration measurements are not necessary!*

*So, in general, it remained quite difficult for me to come to a decision here due to the lack of any quality control for the CH4 concentration data obtained by the two HydroC CH4 sensors (missing post/intermittent calibration or verification during a one-year deployment of in situ sensors!). However, I agree that this study found significantly larger fluctuations in the CH4 concentrations than can be expected from the sensor and, in addition, this study is rather focusing on relative values (methane variability and its driving factors) than on absolute values (calculation of methane fluxes and inventories). Therefore, I have decided to accept most changes in the manuscript provided by the authors - but I still have the following requests for minor changes in the manuscript:*

*• The material & methods section should be extended by a sentence indicating that no post and/or intermittent calibration or validation of the sensors was performed. In addition, a further section in the manuscript should mention that the deployed sensors do have strong limitations for providing quantitative measurement, but that, however, the obtained CH4 data should be sufficient to support the scientific results in this specific manuscript (e.g. something similar to your statement „Even though modern HydroC CH4s with our pump/membrane/detector setup can give decent accuracy and be applicable in a wide range of settings, it might still be relevant to acknowledge that the analysis and discussions in the current manuscript mainly concerns large changes and high concentrations. Considering this, we believe the response time corrected Contros HydroC CH4 data should be more than sufficient to support the scientific results and inferences described in our manuscript").*

The HydroC CH$_4$ sensor deployed in our study is a different sensor in several crucial aspects compared to the older NDIR systems. These changes apply directly to the ability of the sensor to provide quantitative measurements and long-term stability. In our opinion, it is misleading to state that the sensor has strong limitations in providing quantitative measurements as long as the long response time due to the membrane exchange process is

corrected for (which is the case in the present study, using the method detailed in Dølven et al., 2021). This procedure also lets us explicitly report the data accuracy (Fig. B1b), something which was in practice impossible in the past due to the long response time, thereby also conveying explicit probabilistic (confidence intervals) information on the limitations of the data we present.

Evaluation of a measurement should be related to the properties of the phenomenon of interest – some applications require very high accuracy, while others require fast response times, etc. Since we are confident that the data quality is more than sufficient to support our inferences, we find that descriptions of sensor performance should be anchored in reporting our estimated measurement uncertainties, intervals, and general principles etc, rather than stating that the sensor has strong limitations for providing quantitative measurements (for certain applications it probably has - most sensors have applications where they fall short).

That being said, we completely agree with the reviewer that verification of scientific data is essential and long-term deployments are especially challenging – this applies to all deployed sensors. Nonetheless, some sensors are more prone to drifts and erratic behavior than others and systems based on NDIR have a wide range of challenges, especially in long-term deployments with varying environmental parameters. But many of these challenges are alone overcome by using a TDLAS detector (I lack a full account of the various HydroC revisions, but I know other improvements have also been implemented, such as the pump system). Below I elaborate on some of the crucial differences between the TDLAS based system and a NDIR based system in long-term deployments which can hopefully clarify some aspects and justify why we find the addition of the suggested statements (on poor long-term stability and inability to provide quantitative measurements) inappropriate to add without further explanations (implemented changes are found in red below).

For reference see e.g. Lackner (2007), Shemshad et al. (2012), Wang et al. (2019), or Zhang et al. (2021), but also technical description of the sensor https://www.4h-jena.de/en/maritime-technologies/sensors/hydrocrch4/.

Quantitative quality of the measurements

As opposed to NDIR, which emits light at a comparatively wide and constant (assuming no drift) frequency band constrained using an interference filter, a TDLAS based system performs its measurement by scanning over the absorption line(s) of the gas of interest using a laser (i.e. very narrowband, near monochromatic, near coherent light) source with tunable output wavelength. This leads to several advantages with regards to quantitative quality over NDIR:

1. Lower detection limit due to a specifically tuned light source (NDIR also selects appropriate wavelengths but uses an interference filter which usually has a much cruder spectral resolution.)
2. Much less susceptibility to pressure broadening effects (in NDIR the broad band results in pressure modulation of the signal due to changes in the absorption line surface)

3. Overlapping absorption bands are a challenge with NDIR technology because of the constant and wider (and non-discriminatory) frequency band. This is not an issue with TDLAS due to the very narrow and tunable frequency band which makes it possible to explicitly identify where absorption bands of different gases interlace and thus obtain a clear separation between different gases.

All these aspects increase the quantitative quality of the data.

Long-term stability/calibration

NDIR systems are often challenged by poor long-term stability due to several effects such as variations/degradations in the light source over time (both causing spectral and intensity shifts), degradation of detector and contaminations in the measuring chamber. This makes intermittent calibration by e.g. zero gas crucial to monitor and compensate for long-term stability in NDIR based systems (such as the zeroing technique in the Contros HydroC $CO_2$, see Fitzek, 2014). This is not necessary in the same way for TDLAS based systems because the frequency (spectral band) of the light source (laser) is controlled, which not only removes the issue of drift in the spectral band, but also makes it possible to compensate/check for drift by tuning the laser between non-absorbing and $CH_4$ absorbing wavelengths. The TDLAS detector in the Contros HydroC $CH_4$ does this. Although we agree that intermittent calibration would be advantageous for all sensors in such a long-term deployment, this is not specifically necessary for the TDLAS HydroC system (but it would be, for a NDIR-based system).

The main challenge with quantitative quality provided by TDLAS based sensors deployed in dynamic domains is their long response time, which alone can easily result in large discrepancies with other methods and result in raw data which does not represent reality. However, we correct for this by doing a response time correction of the signal. Furthermore, we found no suspicious behavior in neither data nor meta data (i.e. internal temperature, pressure, etc.) during post-processing or in the evaluation of model fit residuals in the response time correction procedure (which modulated the signal variance and was otherwise Gaussian - as expected). All this considered, we believe that the quantitative quality of the measurements should be more than sufficient in our application. We also cannot find concrete evidence to support, nor a relevant reason to add a sentence stating that that the sensor has strong limitations when it comes to providing quantitative measurements in the methods section of this manuscript.

Nonetheless, we agree with the reviewer that it is important to clarify these aspects in the method section and have attempted to incorporate the suggested changes ("*The material & methods section should be extended by a sentence indicating that no post and/or intermittent calibration or validation of the sensors was performed. In addition, a further section in the manuscript should mention that the deployed sensors do have strong limitations for providing quantitative measurement*"), but at the same time reflect what we iterated above by rewriting/adding the following text in the method section (ln 96-104):

Uncertainty ranges for the $CH_4$ sensor data are reported as 95% confidence intervals and typically vary between 5 and 20% (Figure B1b). We did no post and/or intermittent validation. Although

always an advantage for all sensors in long-term deployments, this is not a requirement for the TDLAS based sensor (as opposed to NDIR), due to its high long-term stability. Standard post-processing (e.g. inspection of meta data such as internal pressure and temperature) and evaluation of fit residuals in the response time correction procedure (see Appendix B and Dølven et al. (2021)) also indicated consistent sensor behavior. It is also worth noting that the current manuscript concerns large changes and high concentrations and we are confident that the quality of the response time corrected Contros HydroC CH4 is sufficient to support the inferences described herein.

We also added a description of the error fit residuals in Appendix B (ln 416-419):

Inspection of model fit residuals showed a slight modulation following the variance in the signal, explained by our choice to use the same 3-minute measurement grid across a relatively wide variance ranges, but were otherwise Gaussian. Although expected, this indicates that errors might be slightly overestimated for low-variance sections of the time-series and vice versa for high-variance sections.

And clarified which version of the sensor we deployed (ln 85-88):

The deployed HydroC CH4, being a younger iteration of the sensor, rely on a Tunable Diode Laser Absorption Spectrometry (TDLAS) detector (rather than non-dispersive infrared spectrometry (NDIR)), while the CO2 sensors use NDIR detectors.

*• The authors did not follow my suggestion to implement some basic calculations of CH4 inventories but, however, they came up with an alternative approach to emphasize the urgent need of continuous measurements to detect the temporal variability of seabed seepages. I like the idea of calculating the expected error from unresolved short-term variability for a hypothetical discrete water sampling survey seeking to estimate seep site averages, the newly added section (line 204-242) and its corresponding figure (Figure 4). Note: Correct the typo in the y-axis label and consider using a colorblind-friendly palette*

Thank you for acknowledging this and for suggesting a deeper analysis on what these measurements mean for budget estimates. We fixed the typo and changed the color palette which should now be colorblind friendly.

*• There are still various fragments of grids and box lines of the individual subfigures (mostly grey lines) in Figure 2, but I assume (and hope) this will be corrected before publication. Also accounts for Figure 6.*

This should now be fixed. The problem apparently arose in certain pdf readers and was dependent on the zoom level on the screen.

*Several dois are still not correct formatted, e.g. Silyakova et al. (2020): "https://doi.org/https://doi.org/10.1016/j.csr.2019.104030" – remove the duplicate "https://doi.org/" and also check all other references!*

The duplicates were removed and the reference list has been checked. I apologize for not noticing this error in the first round of review where it was already commented on.

**Response to editor**

Good to hear. We added a short paragraph regarding the calibration/validation of our HydroC CH4 sensor in the methods section - see reply to review above regarding this issue.

L23: We changed y to yr

L35: There was indeed an issue with this sentence so we rewrote it.

L36: we modified the sentence as suggested

L74-L75: The references used here were indeed not consistent with each other. We changed the reference and now use more "classical" values for the region. These values are all in practical salinity.

Table1 & Table 2: we added a line including units below the parameters.

L185: Done

L252: We agree that the use of z-score might be confusing, so we rewrote this part of the text (in addition to in the figure).

L268: Done

L287-288: We removed all water masses abbreviations in the text to make it easier to read.

L293: We removed all abbreviations for Prins Karls Forland

L301: Done

L487: I added the page number and doi to this reference. Was this what was missing?

L500: We added a link to the data-sheet pdf (same iteration of the sensor that we used), since the manual is not available.

L503: Corrected

L510: This manuscript is still under open discussion. We received one favorable review (https://doi.org/10.5194/gi-2021-28-RC1), but are still missing one review.

**Other changes**

We changed the ordering of some of the paragraphs in the methods section to better fit the addition above and made minor changes at a few places during a final read through. All is shown in the marked-up version.

**References**

Dølven, K. O., Vierinen, J., Grilli, R., Triest, J., and Ferré, B.: Response time correction of slow response sensor data by deconvolution of the growth-law equation*, Geoscientific Instrumentation, Methods and Data Systems Discussions [preprint]*, 1–22, https://doi.org/10.5194/gi-2021-28, in review, 2021.

Shemshad, J., Aminossadati, S. M., Kizil, M. S.,: A review of developments in near infrared methane detection based on tunable diode laser, *Sensors and Actuators B: Chemical*, 171-172, 77-92, 2012. *https://doi.org/10.1016/j.snb.2012.06.018*

Wang, F., Jia, S., Wang, Y. and Tang, Z.: Recent Developments in Modulation Spectroscopy for Methane Detection Based on Tunable Diode Laser, *Applied Sciences*, 9(14), 2816, doi:10.3390/app9142816, 2019.

Schiff, H.I., Mackay, G.I. & Bechara, J. The use of tunable diode laser absorption spectroscopy for atmospheric measurements. *Res Chem Intermed*, 20, 525–556, https://doi.org/10.1163/156856794X00441, 1994.

Lackner, Maximilian. Tunable diode laser spectroscopy in the process industries: a review. *Reviews in chemical engineering*. 23. 65, https://doi.org/10.1515/REVCE.2007.23.2.65, 2007.

Zhang, Z., Li, M., Guo, J., Du, B. and Zheng, R.: A Portable Tunable Diode Laser Absorption Spectroscopy System for Dissolved $CO_2$ Detection Using a High-Efficiency Headspace Equilibrator. *Sensors (Basel, Switzerland)*, *21*(5), 1723, https://doi.org/10.3390/s21051723, 2021.

Fietzek, P., Fiedler, B., Steinhoff, T., & Körtzinger, A.:  In situ Quality Assessment of a Novel Underwater pCO2 Sensor Based on Membrane Equilibration and NDIR Spectrometry, *Journal of Atmospheric and Oceanic Technology*, 31(1), 181-196. ,2014.